# Uniform bacterial genetic diversity along the gut

Michael Wasney [1,8], Leah Briscoe [2,8], Richard Wolff[3], Hans Ghezzi[4], Carolina Tropini [5,6,7] & Nandita Garud [1,2,3] ✉

While environmental gradients are known to result in heterogeneous distributions of bacterial species along the gastrointestinal tract, the spatial distribution of genetic diversity within these species remains poorly understood. Because bacterial genetic variants influence host traits like inflammation and metabolism, understanding their distribution is critical. Here, we analyze ~30 common gut commensals in germ-free mice colonized with the same healthy human stool. Unexpectedly, we find that while species composition varied significantly across gut regions, genetic diversity within species remained remarkably uniform. This uniformity is driven by similar strain frequencies along the gut lumen, indicating that genetically divergent strains can coexist without spatial segregation. Furthermore, ~60 evolutionary adaptations arising within the mice tend to sweep globally throughout the gut, showing little region-specificity. We observe similar dynamics in conventional mice and humans, suggesting that uniform bacterial genetic diversity is a conserved, robust feature of mammalian gut ecosystems.

The spatial organization of microbiota along the gut plays a critical role in microbiome function[1]. Previous research in humans and mice has established that different regions of the gut, such as the small and large intestines, harbor distinct assemblages of bacterial species[2,3], and that disruption of this organization is associated with disease[4]. Spatial organization in the microbiome is thought to arise from distinct niches imposed by environmental and physiological gradients along the gut[3,5,6]. However, it remains unknown whether such gradients drive similar spatial organization of within-species bacterial genetic diversity, such as by determining the occupancy of genetically distinct strains or individual genetic variants along the gut. Understanding spatial organization at this level is crucial, as bacterial genetic diversity drives traits that support proper microbiome function and are associated with multiple host-relevant phenotypes[7], including ability to digest food[8], local inflammation in specific regions of the gut[9], antibiotic resistance[10], metabolic capabilities[8,11–14], and pathogen resistance[15].

There are several ecological and evolutionary mechanisms that could result in either (i) spatially segregated or (ii) well-mixed genetic variation along the gut (Fig. 1A). Spatial segregation could arise because environmental gradients select for different strains and mutations along the gut, co-colonizing strains occupy distinct physical niches to reduce competition, or slow migration rates limiting the spread of genetic variation. Contrastingly, spatial uniformity could arise because adaptive variants are globally adaptive along the gut, co-colonizing strains occupy distinct non-physical (e.g., metabolic) niches, or rapid migration rates.

Previous studies have sought to examine the spatial distribution of strains and individual nucleotide variants. One study looking at a small number of commensal bacterial species found that different strains can coexist along the human gut[16], but did not determine whether strain frequencies of these co-colonizing strains were variable in different gut regions or whether coexistence was a general

[1]Human Genetics, University of California, Los Angeles, Los Angeles, CA, USA. [2]Interdepartmental Program in Bioinformatics, University of California, Los Angeles, Los Angeles, CA, USA. [3]Ecology and Evolutionary Biology, University of California, Los Angeles, Los Angeles, CA, USA. [4]Department of Bioinformatics, University of British Columbia, Vancouver, BC, Canada. [5]Department of Microbiology and Immunology, University of British Columbia, Vancouver, BC, Canada. [6]School of Biomedical Engineering, University of British Columbia, Vancouver, BC, Canada. [7]Humans and the Microbiome Program, Canadian Institute for Advanced Research, Toronto, ON, Canada. [8]These authors contributed equally: Michael Wasney, Leah Briscoe. ✉e-mail: ngarud@ucla.edu

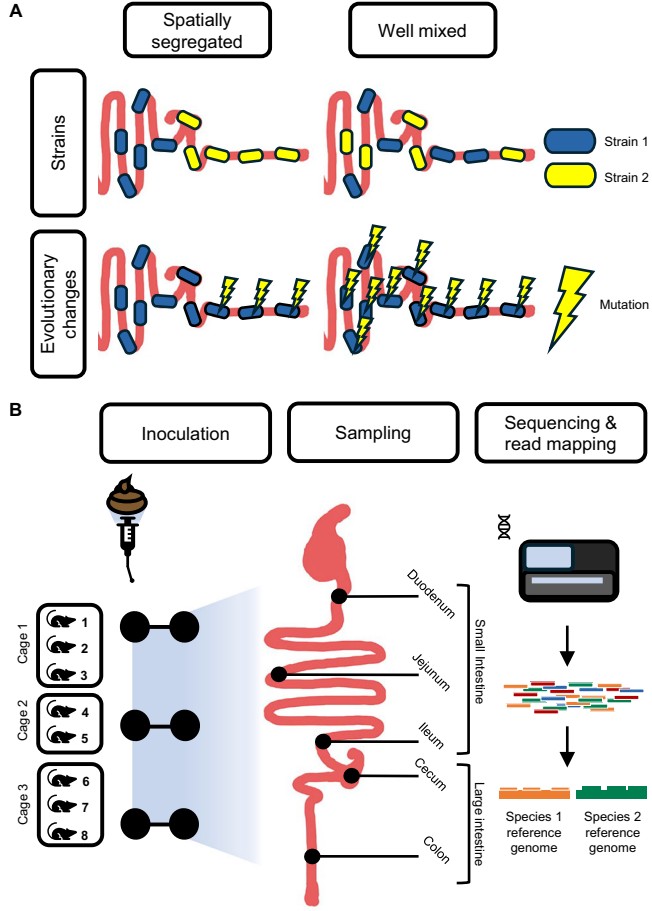

**Fig. 1 | Hypotheses and experimental design. A** Possible organization of genetic diversity along the gut. Strains and their evolutionary changes may be either spatially segregated, or evenly mixed. **B** Experimental design. Eight germ-free Swiss-Webster mice were inoculated with a single healthy human-derived stool sample[3]. Subsequently, the microbiota was allowed to equilibrate over eight weeks before shotgun sequencing was performed on samples collected from five regions along the gut (small intestine: duodenum, jejunum, and ileum; large intestine: cecum and colon). The duodenum, jejunum, and ileum are considered regions of the small intestine, and cecum and colon are considered regions of the large intestine. To quantify species abundances and SNV frequencies, metagenomic reads were mapped against a reference database of genomes from almost 6000 bacterial species.

phenomenon. Other recent work found that single nucleotide variants (SNVs) can arise locally along the gut for a few bacterial species[16,17], though these studies did not demonstrate that spatially specific evolutionary changes were adaptive or common across bacterial species or host. In another case, a uropathogenic strain of *Enterococcus gallinarum* evolved adaptive mutations specific to the mucosa, which caused inflammation[9]. However, it is unclear how commonly adaptive mutations spatially segregate along the longitudinal axis of the gastrointestinal (GI) tract (i.e., from small to large intestine), particularly among commensal microbes not facing any strong selective perturbations such as antibiotics. Here we will test the hypothesis that strains and their adaptations may also display spatial segregation, similar to the spatial heterogeneity observed at the species level.

To date, characterizing the spatial structure of genetic diversity along the mammalian gut has been challenging because of the invasive nature of directly sampling the GI tract, the limited bacterial biomass obtained from such attempts, and because of the difficulties in culturing a broad diversity of taxa found natively in the human gut microbiome. However, these challenges can be circumvented using a

combination of experimental and statistical approaches. First, the use of a gnotobiotic mouse model for studying the gut microbiome enables the study of human gut-specific species of bacteria in vivo in a controlled manner as a single inoculum can be administered to multiple mice, allowing us to measure variability across replicates[18–21]. Second, the use of mice overcomes the hurdle of invasive sampling in humans as the full gut contents can be extracted after sacrifice and then processed for sequencing. Third, the use of shotgun metagenomic sequencing enables the characterization of genetic diversity across a broad panel of bacterial taxa, including those undetectable using culture-based methods[22]. Finally, recent statistical innovations in phasing genotypes of individual strains from shotgun metagenomic data enables the ability to quantify strain frequencies and their evolutionary modifications[23–25].

In this study, we capitalize on these advances to assess the distribution of strains and evolutionary modifications along the gut. Specifically, we analyzed the frequencies of bacterial strains and the evolutionary modifications they carry within luminal samples obtained from along the GI tracts of germ-free mice eight weeks after they were inoculated with the microbiome of a single healthy human (Fig. 1B). We previously analyzed this same humanized mouse dataset with 16S rRNA sequencing of luminal contents and found significant shifts in abundance of taxonomic family members along the gut[3]. By using metagenomic sequencing to analyze the same samples at a deeper level, we were able to estimate strain frequencies and identify evolutionary modifications. We report that the frequency of strains and evolutionary modifications found in most bacterial species was unexpectedly uniform across the gut. Confirming that this uniformity extends to other natural microbiomes, we applied the same analyses to (i) a cohort of conventional mice with their own naturally acquired, host-adapted microbiomes and (ii) a cohort of healthy humans whose luminal contents from multiple gut regions were sampled using ingestible capsule devices swallowed over several days. In all systems, we observed spatial uniformity of genetic diversity along the GI tract, suggesting that this pattern may be a common feature of multiple mammalian gut microbiomes. These results further imply that species abundance differences along the gut may play a greater role than genetic differences in responding to environmental gradients along the gut, and that this could be a robust feature of gut microbial ecology across both mouse and human hosts.

## Results

To investigate the effect of gut region on microbial strain diversity, we performed shotgun sequencing on microbiomes from eight mice that were orally gavaged germ-free Swiss-Webster mice with a single stool sample (hereafter referred to as the inoculum) from a healthy human donor in a previous study[3] (Fig. 1B). Luminal contents were collected from five intestinal regions in each mouse: the duodenum, jejunum, ileum, cecum, and colon. DNA was extracted and metagenomic sequencing was performed on each of the five mouse intestinal samples as well as the original human inoculum (Supplementary Data 1, "Methods").

To infer species abundances and SNV frequencies, we mapped shotgun reads to a database of reference genomes[26] for the most prevalent and abundant bacterial species present in these mice (Fig. 1B) (see "Methods" for more details on the bioinformatic pipeline used). Summaries of species and genetic-level diversity have been previously quantified from stool with similar approaches (e.g., Nayfach et al.)[26], but here we investigate this diversity along the gut.

### Bacterial species and functional diversity differ along the gut

First, we examined whether bacterial species diversity, measured using alpha diversity (the Shannon index), varied along the tract of the gut. We hypothesized the large intestine would have higher alpha diversity compared to other regions of the gut, as previous studies in humans[27]

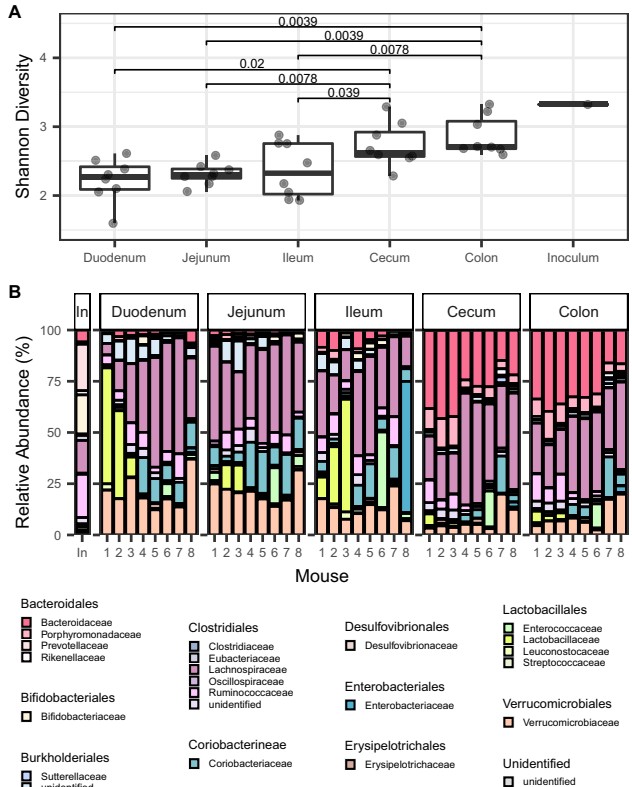

**Fig. 2 | Increase in taxonomic diversity and change in community membership along the length of the gut. A** Alpha diversity (Shannon Index) estimates for bacterial species in the inoculum ($n = 1$) and different regions of the gut for all mice ($n = 8$). Each point represents the alpha diversity in a mouse. Boxplot center lines represent the median, hinges represent the first and third quartiles, and the whiskers extend to the maximum and minimum point of each distribution. Two-sided Wilcoxon signed-rank tests were performed between all possible within-host pairs of small intestinal versus large intestinal samples ($n = 8$ for each paired test, corresponding to 8 mice). *P*-values for each test are indicated over the brackets. **B** Relative abundance of bacterial families in the five gut regions and inoculum. Bacterial families are labeled with unique colors and grouped by order in the legend.

and mice[19,20] have found that species diversity is highest in the colon due to lower flow rates and the relatively high abundance of host- and food-derived complex polysaccharides[5,6,28]. As expected, gut region was a major driver of alpha diversity, with the large intestine having significantly higher average alpha diversity than the small intestine (fold change in average diversity between large and small intestine per mouse = 1.22; two-sided Wilcoxon signed-rank test, $p = 0.0156$) (Fig. 2A).

Next, we hypothesized that the relative abundance of individual bacterial community members differ along the gut, as factors such as oxygen availability, antimicrobial peptides, and carbon-source favor different dominant taxa in the mammalian small and large intestines, respectively[1,2,29,30]. To quantify shifts in community composition along the gut, we examined the relative abundance of bacterial families, expecting to see saccharolytic, obligate anaerobic taxa dominate in the large intestine and rapidly dividing facultative anaerobes dominate in the small intestine[2,6,26,28,29]. Supporting our prediction, several families in the order Bacteroidales−an order of anaerobic bacteria comprising many species which are involved in breaking down complex sugars in the colon[31]−were enriched in the large intestine (Fig. 2B and Supplementary Fig. 1), including the families Bacteroidaceae (median log2 fold change per mouse = 0.338; two-sided Wilcoxon signed-rank test,

mouse = $8.58 \times 10^{-2}$; two-sided Wilcoxon signed-rank test, $p = 0.0143$), and Rikenellaceae (median log2 fold change per mouse = $6.23 \times 10^{-3}$; two-sided Wilcoxon signed-rank test, $p = 0.0209$). Meanwhile, the family Lactobacillaceae, which is known to colonize the human small intestine with the aid of unique mucus-binding proteins (MUBs) which adhere to the small intestinal mucosal layer[29], was enriched in the small intestine relative to the large intestine (median log2 fold change per mouse = −0.121; two-sided Wilcoxon signed-rank test, $p = 0.0225$) (Supplementary Fig. 1). These enrichments in the small intestine confirm that despite human stool more closely resembling the community composition of the human large intestine[30], inoculation with a human fecal sample is sufficient to support colonization by small intestine-associated taxa. Together, these results are in line with previous work showing that the composition of microbial taxa differs along the gut[3,28].

Lastly, to investigate community-wide functional diversity along the gut, using HUMAnN3[32], we mapped metagenomic reads to gene families and MetaCyc pathways. Across all samples, HUMAnN3 identified 808,204 gene families corresponding to 470 MetaCyc pathways, which were subsequently aggregated at the community level by summing read counts across taxa. These pathway profiles were then analyzed using MaAsLin3[33] to test for differences between the small intestine and large intestine. The model incorporated gut location, cage, and read count as fixed effects and used subject-level stratification for paired comparisons−revealing 80 pathways significantly associated with gut location ($q < 0.05$). Among these, 33 pathways showed highly significant associations ($q < 0.001$), and their coefficients are shown in Supplementary Fig. 2 (Supplementary Data 2). The same families we found to be enriched in the large intestine were also the source of many of the pathways most significantly associated with the large intestine, namely Bacteroidaceae (25 pathways) and Rikenellaceae (14 pathways). Meanwhile, a family enriched in the small intestine, Lactobacillaceae, was the source of 7 pathways significantly associated with the small intestine.

## Bacterial genetic diversity along the gut

Having confirmed that taxonomic diversity varies spatially along the gut, we next investigated whether the same was true of within-species genetic diversity. First, we quantified the distribution of nucleotide diversity ($\pi$) for the 30 most abundant species detected in the inoculum and mice ("Methods") (Supplementary Fig. 3 and Supplementary Data 3). Across these species, nucleotide diversity measured in the inoculum was highly variable, ranging from $2.2 \times 10^{-4}$ in *Odoribacter splanchnicus* to $1.2 \times 10^{-2}$ in *Bacteroides vulgatus*, spanning two orders of magnitude (Fig. 3A). High values of $\pi$ ($> 10^{-3}$) are generally inconsistent with the colonization and subsequent diversification of a single strain during a host's lifetime, and instead support the presence of two or more strains that diverged long before colonization[23]. Meanwhile, low values of $\pi$ are more consistent with colonization by a single strain[23]. Thus, the large range of $\pi$ values observed across species indicates that in some instances, there are multiple co-colonizing strains of the same species, and in other instances, there is only one strain.

Average nucleotide diversity values in the mice were positively correlated with nucleotide diversity in the inoculum (Spearman $\rho = 0.845, p = 7.55 \times 10^{-7}$) (Fig. 3A and Supplementary Fig. 4). However, despite this overall similarity, a range of nucleotide diversity values was observed within individual mice, with some mice having levels comparable to those seen in the inoculum and others having substantially lower values, suggesting variation in lineage structure across mice. Specifically, for the species with the highest values of nucleotide diversity in the inoculum ($\pi > 1 \times 10^{-3}$), values across mice were often bimodally distributed. For example, $\pi$ values for *Bacteroides vulgatus* were either all clustered around $10^{-2}$ in some mice or around $10^{-3}$ and lower for other mice (Fig. 3A). This variation suggests that in some

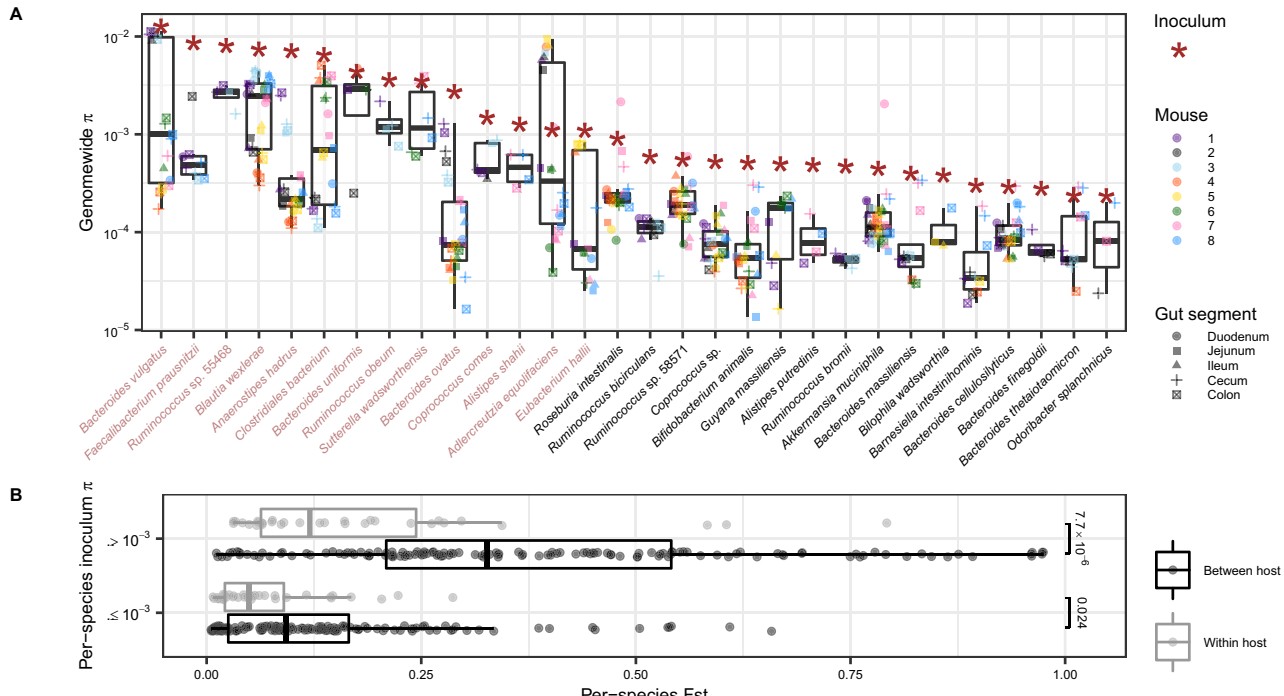

**Fig. 3 | Genetic diversity measured in 30 gut commensal species. A** Nucleotide diversity ($\pi$) estimates for the 30 most abundant and prevalent species ("Methods"). Each dot represents nucleotide diversity ($\pi$) in a specific gut region of an individual mouse. Dot color corresponds to the mouse, while dot shape indicates the gut region. Red asterisks denote nucleotide diversity measured in the inoculum for that species. Species names in light red correspond to those species with $\pi > 1 \times 10^{-3}$ in the inoculum. All nucleotide diversity estimates as well as the number of estimates for each species can be found in Supplementary Data 3. **B** Fixation index ($F_{ST}$) between all pairs of mouse samples for the 30 species analyzed in (**A**). Each data point represents a single species being compared between two samples, with points colored according to whether the sample comparison is within host (grey) or between hosts (black). Data points are further separated based on whether the species had $\pi > 10^{-3}$ or $\pi \leq 10^{-3}$ in the inoculum ($\pi > 10^{-3}$, between host pairs $n = 136$; $\pi > 10^{-3}$, within host pairs $n = 32$; $\pi \leq 10^{-3}$, between host pairs $n = 146$; $\pi \leq 10^{-3}$, within host pairs $n = 34$). Two separate two-sided Wilcoxon rank sum tests were performed comparing the distributions of $F_{ST}$ within versus between hosts for species with $\pi > 10^{-3}$ and $\pi \leq 10^{-3}$ in the inoculum, with the *p*-value for each test displayed directly to the right of the distributions being compared. In both (**A. B**), boxplot center lines represent the median, hinges represent the first and third quartiles, and the whiskers extend to the maximum and minimum point of each distribution. $F_{ST}$ estimates for all pairs considered here can be found in Supplementary Data 4.

mice, multiple strains colonized, and in other mice, a single strain colonized even when multiple were present in the inoculum. However, $\pi$ never spanned an order of magnitude within a single mouse, potentially implying more similar strain colonization dynamics along the gut within mice than across mice.

To understand if nucleotide diversity differed between regions of the gut, we measured $F_{ST}$, or fixation index, between the small and large intestine for each species. $F_{ST}$ is a distance metric where high values indicate large genetic differentiation between two samples[34]. We hypothesized that $F_{ST}$ would be elevated between gut regions within hosts, either due to evolutionary changes accruing or variable strain frequencies. Contrary to this expectation, we observed that $F_{ST}$ between the small and large intestine of the same host was on average low for the 30 species examined (0.119), especially relative to the average $F_{ST}$ between mice at the same gut region (0.225) (Fig. 3B and Supplementary Data 4). This difference was more pronounced in species with high nucleotide diversity in the inoculum ($\pi > 10^{-3}$) (two-sided Wilcoxon rank sum test, $p = 7.7 \times 10^{-6}$) compared to species with low diversity ($\pi \leq 10^{-3}$) (two-sided Wilcoxon rank sum test, $p = 0.024$).

Species with high nucleotide diversity in the inoculum may have larger differences in $F_{ST}$ within versus between mice than species with low nucleotide diversity in the inoculum because the former exhibits more variable strain colonization dynamics. Specifically, when multiple strains are present in the inoculum, there is a potential for different combinations of strains to colonize different mice, leading to higher $F_{ST}$ and differences in $\pi$ between mice. Within mice, $F_{ST}$ tends to be low

and $\pi$ tends to be uniform for these species, suggesting that similar combinations of strains are found all along the gut tract of the same host. Meanwhile, when there is only one strain present in the inoculum, the same strain should colonize all mice and gut regions. This is consistent with the observed low values of $F_{ST}$ and more uniform values of $\pi$ both within and between mice for these latter species. In the next section, we further investigate whether similar combinations of strains are found along the gut tract when nucleotide diversity is high.

## Strain frequencies are uniform along the gut within hosts

We next asked whether the identities and frequencies of strains colonizing different parts of the gut of a given mouse are the same or different when multiple strains are present in the inoculum. Variation in occupancy or frequency along the gut would suggest that physical niche partitioning is important in maintaining multiple conspecific strains in the same host that might otherwise compete in the same physical niche, or that distinct environmental pressures along the gut select for different strains.

To understand whether frequencies of co-colonizing strains of the same species vary along the gut, we inferred strain frequencies for the 14 species with levels of nucleotide diversity $> 10^{-3}$ in the inoculum, as these samples had sufficiently high enough diversity to potentially harbor at least two strains (Fig. 3). To infer strain frequencies, we used an algorithm previously applied to metagenomic time series data[24,25] ("Methods"). For the species *Alistipes shahii*, *Bacteroides ovatus*, *Coprococcus comes*, and *Faecalibacterium prausnitzii*, only one strain

was inferred to be present within the mouse samples. For the rest of the species, we inferred that two strains were present (Fig. 3, Supplementary Fig. 5, and Supplementary Data 5). We next asked whether the frequencies of strains for species with two strains were uniform or variable along the gut.

Application of the strain phasing algorithm revealed that frequencies of co-colonizing strains were relatively uniform across gut regions within hosts for most bacterial species and mice, contrary to the expectation that their frequencies may differ due to niche partitioning or environmental selection. For example, for the species *Blautia wexlerae*, the two strains were able to coexist at roughly 40:60% frequency along all five regions of the gut in mouse 8, and for the species *Adlercreutzia equilofaciens*, the two strains were able to coexist at roughly 20:80% frequency in all parts of the gut in mice 4 and 5 (Fig. 4A). To understand whether co-colonizing strains generally displayed uniform frequencies along the gut, we calculated change in major strain frequency ($\Delta f$) between mouse samples for 7 species that were present in multiple mice, cages, and gut regions ("Methods"). Across the seven species, median change in frequency between gut regions within a host was only $9.36 \times 10^{-3}$, much smaller than the median change in frequency we observed across mice within the same cage ($3.49 \times 10^{-2}$) or between mice in different cage ($3.09 \times 10^{-1}$) (Fig. 4B and Supplementary Data 6).

To confirm that strain frequencies were generally more homogeneous within versus between hosts for all species in our dataset, we performed a series of ANOVA tests to estimate the proportion of variance in major strain frequencies attributable to gut region ("gut region"), mouse identity ("mouse"), and cage identity ("cage") (Supplementary Fig. 6). Consistent with the observations of uniformity in genetic variation along the gut, gut region explained the lowest variance for all seven bacterial species (average variance explained = 2.70%), suggesting that gut region is less important in explaining strain frequencies than mouse identity or cage identity. By contrast, mouse and cage explained substantially greater variation (average combined variance explained = 90.0%). These observations support the idea that strain frequencies along the gut were generally uniform across species and reveal that multiple strains were able to coexist in the same region of the gut. Furthermore, they imply some ecological mechanism other than spatial segregation must be responsible for permitting strain coexistence within the same host.

Social transmission of bacteria, through behaviors like coprophagy[14,35], may also facilitate between-host spread of strains, resulting in similar strain frequencies between mice in the same cage. Supporting this notion, strain frequencies were more similar among cohoused mice compared to mice housed in different cages, indicating that migration of strains between mice plays a role in homogenizing frequencies of strains. For example, for the species *A. equilofaciens*, which was present in multiple mice and cages, the within-cage variance of strain frequencies measured in the jejunum ranged from $1.65 \times 10^{-6}$ in cage 3 to $1.29 \times 10^{-3}$ in cage 2, whereas the between cage variance was at least two orders of magnitude greater, $1.18 \times 10^{-1}$ (Supplementary Fig. 7 and Supplementary Data 7). These results align with previous studies that have found coprophagic behavior in mice can facilitate the spread of strains and adaptive variants in *Bacteroides thetaiotaomicron*[36] and *Escherichia coli*[14,35] between co-housed mice, enforcing similar strain frequencies between mice[14]. However, not all species have similar within-cage strain frequencies (e.g., *B. uniformis*, Fig. 4), suggesting barriers to between-host migration in these cases. This species-specific migration effect is reflected in the fact that "cage" explains most of the variance in strain frequency for four species, but very little variance in the remaining three species (Supplementary Fig. 6).

Together, these results indicate that co-colonizing strains do not need to occupy physically distinct niches to colonize the same host. To the contrary, strain frequencies tended to be relatively uniform along the gut. This uniformity may be driven by rapid migration rates between gut regions, which also likely results in greater similarity in strain frequencies among mice of the same cage versus different cages due to social transmission.

## Evolutionary changes are not localized and spread throughout the gut

Even if bacterial strain frequencies are relatively homogenous along the gut, it is possible that individual genetic variants within species are locally restricted along the gut, potentially due to local adaptation. To assess the possibility of local adaptation, we asked if any SNVs have extreme frequency differences between gut regions, or if they too are homogeneously distributed.

SNV frequency changes between gut regions could arise within a host as a consequence of (i) evolutionary changes (e.g., adaptation), (ii) fluctuations in frequencies of genetically distinct strains, or (iii) sampling error. Here, we were interested in distinguishing SNV frequency changes arising as a consequence of evolution from those arising from strain fluctuations or sampling error. To distinguish SNV frequency changes arising due to evolution as opposed to strain fluctuations, we inferred the genotype of one of the strains present in the sample, or "quasi-phased" the genotype[23] and then tracked SNV frequency changes on the background of that genotype between pairs of samples. Doing so allowed us to detect new mutations modifying a resident lineage as opposed to alleles rising to high frequency due to strain fluctuations. To distinguish SNV frequency changes due to evolution (e.g., adaptation) versus sampling error, we considered only large allele frequency changes ($f \leq 0.2$ in one sample to $f \geq 0.8$ in another sample) between samples with a median read coverage of 20 or more and at sites with a minimum read coverage of 20 or more in both samples, which are unlikely to arise due to sampling error or drift[23,37]. Previously, we showed that the genotype of the dominant lineage in samples with a single, high frequency strain can be confidently quasi-phased[23], and that evolutionary changes with large allele frequency changes can subsequently be detected ("Methods").

Using this method, we identified 366 quasi-phaseable genomes from 62 bacterial species present in the intestinal contents of the eight mice and the inoculum (Supplementary Fig. 3). Next, we detected SNV frequency changes between pairs of quasi-phaseable genomes. These pairs were between the inoculum and any mouse (10 bacterial species), different gut regions *within* mice (33 bacterial species), and gut regions *between* mice (28 bacterial species). To avoid comparing distantly diverged lineages that were not descended from the same inoculum strain, we only considered pairs with ≤20 SNV frequency changes. We observed a total of 65 unique SNVs experiencing extreme allele frequency changes across all possible pairs of quasi-phaseable genomes in our dataset (Fig. 5A, Supplementary Figs. 3 and 8, and Supplementary Data 8).

By having an inoculum to compare to, we can ascertain if a lack of SNV differences along the gut is due to the no evolutionary changes accruing, or the global spread of an evolutionary change throughout the gut. We observed 11 SNV frequency changes rising from low frequency in the inoculum to high frequency in mice in 4 species out of the 10 with at least one inoculum-mouse sample quasi-phaseable genome pair (Fig. 5A, Supplementary Fig. 8, and Supplementary Data 8). The rapid timescales over which these SNVs rose to high frequency are consistent with these SNV frequency changes potentially representing adaptations occurring during the host colonization process. Only one of these SNVs, detected in *Coprococcus* sp., showed spatial differentiation in a single host (SNV frequency of 0 in jejunum versus 0.935 in cecum of mouse 1) (Fig. 5A and Supplementary Data 8), indicating that advantageous SNVs tend to spread along the gut. Since not all SNVs observed within mice had sufficient coverage to be analyzed in the inoculum, we next examined SNV frequency changes arising between gut regions. Supporting the observation that SNVs

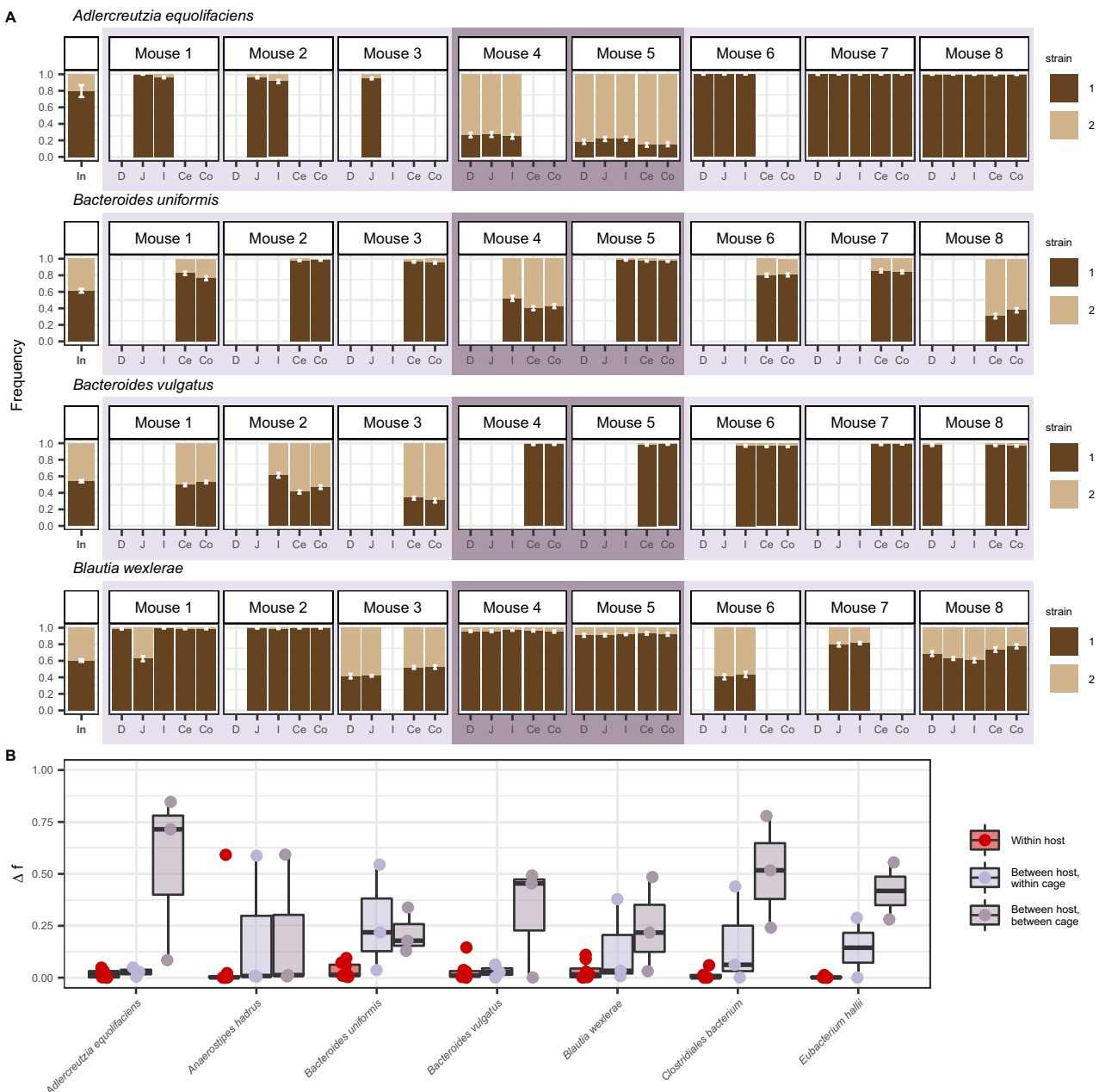

**Fig. 4 | Strain frequencies are similar along the gut but may differ between mice housed in different cages. A** Strain frequencies for four bacterial species detected in the guts of eight mice inoculated with the same human stool sample. The strain frequency in each sample is calculated as the mean of the frequency of SNVs belonging to that strain cluster ("Methods"). Strain 1 and strain 2 frequencies are denoted by dark and light bars, respectively. Error bars represent bootstrapped 95% confidence intervals of the inferred strain frequencies. Mice 1–3 were co-housed in cage 1, mice 4 and 5 in cage 2, and mice 6–8 in cage 3, with the housing scheme indicated by the alternating light and dark purple boxes. Absence of a strain frequency bar in a sample indicates insufficient coverage or species absence ("Methods"). See Supplementary Fig. 5 for the strain frequencies of the ten species with high nucleotide diversity in the inoculum ($\pi > 10^{-3}$) that are not shown here.

Strain frequency values across all samples for species plotted here and in Supplementary Fig. 5 can be found in Supplementary Data 5. **B** For seven species present in multiple mice, cages, and gut regions, change in strain frequency $\Delta f$ was calculated between pairs of samples from within the same host (red), between hosts in the same cage (light purple), and between hosts in the different cages (dark purple). Each black dot represents a different pairwise sample comparison in each species. The pairwise comparisons plotted here have been chosen to reduce redundancy (e.g., only a single within-host $\Delta f$ value is plotted per species per mouse) ("Methods"). See Supplementary Data 6 for precise $\Delta f$ values for all pairs plotted. Boxplot center lines represent the median, hinges represent the first and third quartiles, and the whiskers extend to the maximum and minimum point of each distribution.

spread along the gut, we observed only 4 unique *within-host* SNV frequency changes in 3 out of the 33 species, all between the small intestine and large intestine, including the SNV in *Coprococcus* sp. (Supplementary Data 8). By contrast, we observed 57 unique *between-host* SNV frequency changes in 15 of the 28 species (Fig. 5A, Supplementary Fig. 8, and Supplementary Data 8). The rate of SNV frequency

changes observed between hosts was significantly higher than that within hosts (bootstrapped 95% confidence interval for between-host SNV changes is $1.54 \times 10^{-7}$ differences/bp - $4.42 \times 10^{-7}$ differences/bp; bootstrapped 95% confidence interval for within-host SNV changes is 0 differences/bp - $6.62 \times 10^{-8}$ differences/bp) (Fig. 5B). The low rate of SNV frequency changes within mice relative to between-host and

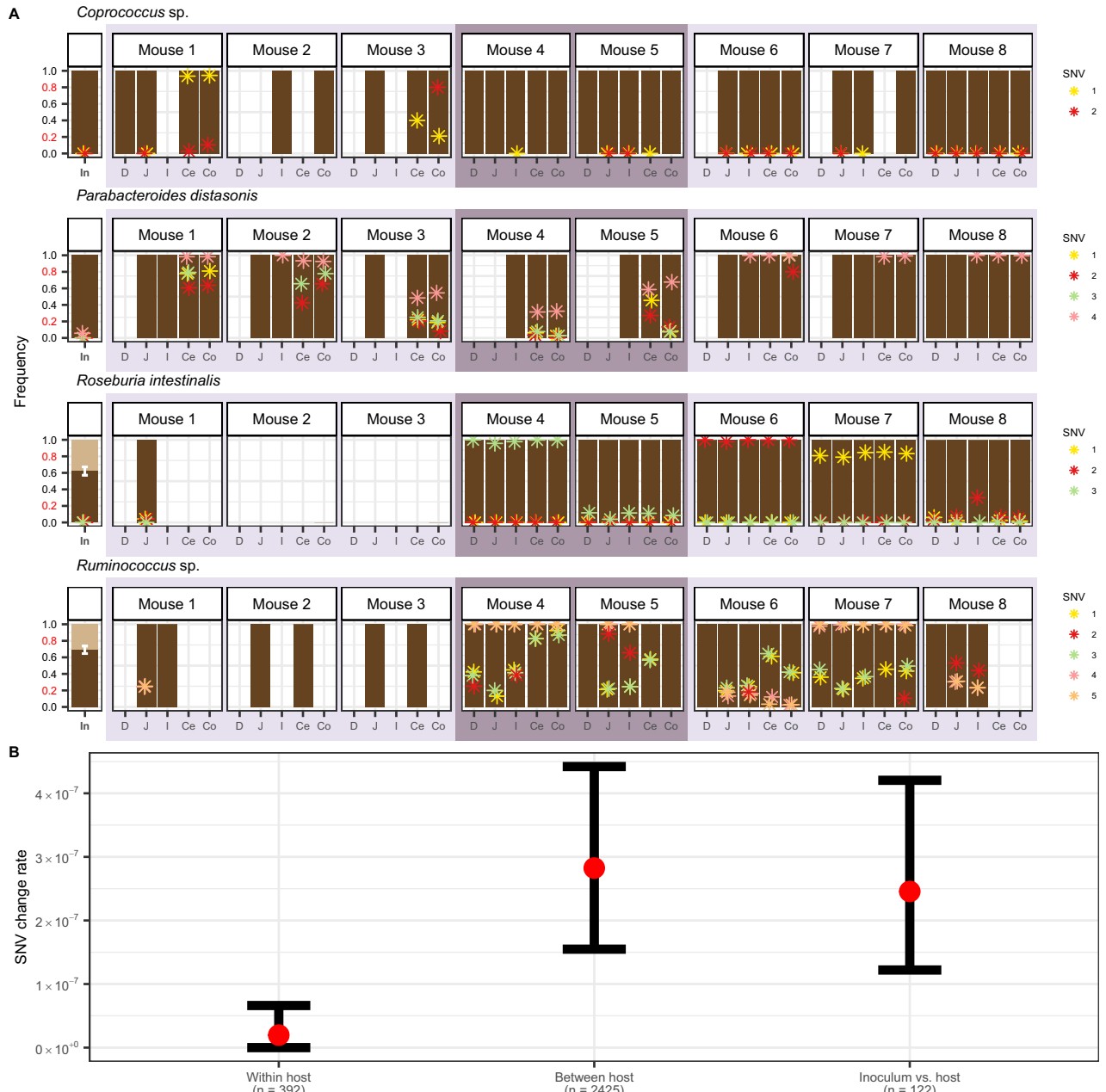

**Fig. 5 | Evolutionary changes are uniformly found along the gut. A** The frequencies of SNVs exhibiting extreme allele frequency changes (allele frequency $f \leq 0.2$ in one sample to $f \geq 0.8$ in another sample) between any pair of samples sharing the same quasi-phaseable strain genome ($\leq 20$ SNV frequency changes) are visualized for *Coprococcus* sp., *P. distasonis*, *R. intestinalis*, and *Ruminococcus* sp. Asterisks represent the allele frequency of a SNV in a given sample, with each SNV within a species having its own unique color. Samples lack asterisks for a particular SNV when those loci do not meet the minimum coverage requirement of 20 reads to infer frequency. When two strains of the same species are present, strain frequencies of co-colonizing strains are represented by dark and light brown bars, respectively, with the error bars around the strain frequency estimate representing bootstrapped 95% confidence intervals of the strain frequency ("Methods"). The remaining 11 species in which SNV frequency differences were detected are shown in Supplementary Fig. 8. The allele frequencies of all 65 SNVs experiencing an extreme allele frequency change (plotted here and in Supplementary Fig. 8) can be found in Supplementary Data 8. **B** Comparison of the rate of SNV frequency changes aggregated across species within mice, between mice, and between inoculum and mice. Colored in red are the observed SNV frequency change rates aggregated across species. Bars represent bootstrapped 95% confidence intervals constructed using the 2.5th and 97.5th percentiles of 1000 bootstrap iterations, with values in each iteration representing the overall SNV allele frequency change rate aggregated across species and sample pairs after randomly sampling 100 quasi-phaseable genome pairs with replacement ("Methods"). The total number of quasi-phaseable genome pairs used to calculate point estimates and bootstrapped confidence intervals of allele frequency change rate are shown below the x-axis label for each category.

inoculum-mouse SNV frequency changes rates suggests that potentially advantageous SNVs overwhelmingly transmit along the gut within hosts.

To understand whether SNVs spreading along the gut showed additional signals of being adaptive beyond extreme allele frequency changes on short time scales, we asked if the same SNV changes arise in multiple mice, as parallelism is a hallmark of adaptation[38–42]. Adaptive variation can arise in multiple mice via independent recurrent de novo mutation or the parallel rise in frequency of standing genetic variation (SGV) already present in the inoculum[42]. Of the 21 unique SNVs that were observed to undergo an extreme allele frequency change between any pair of samples and had sufficient coverage in the

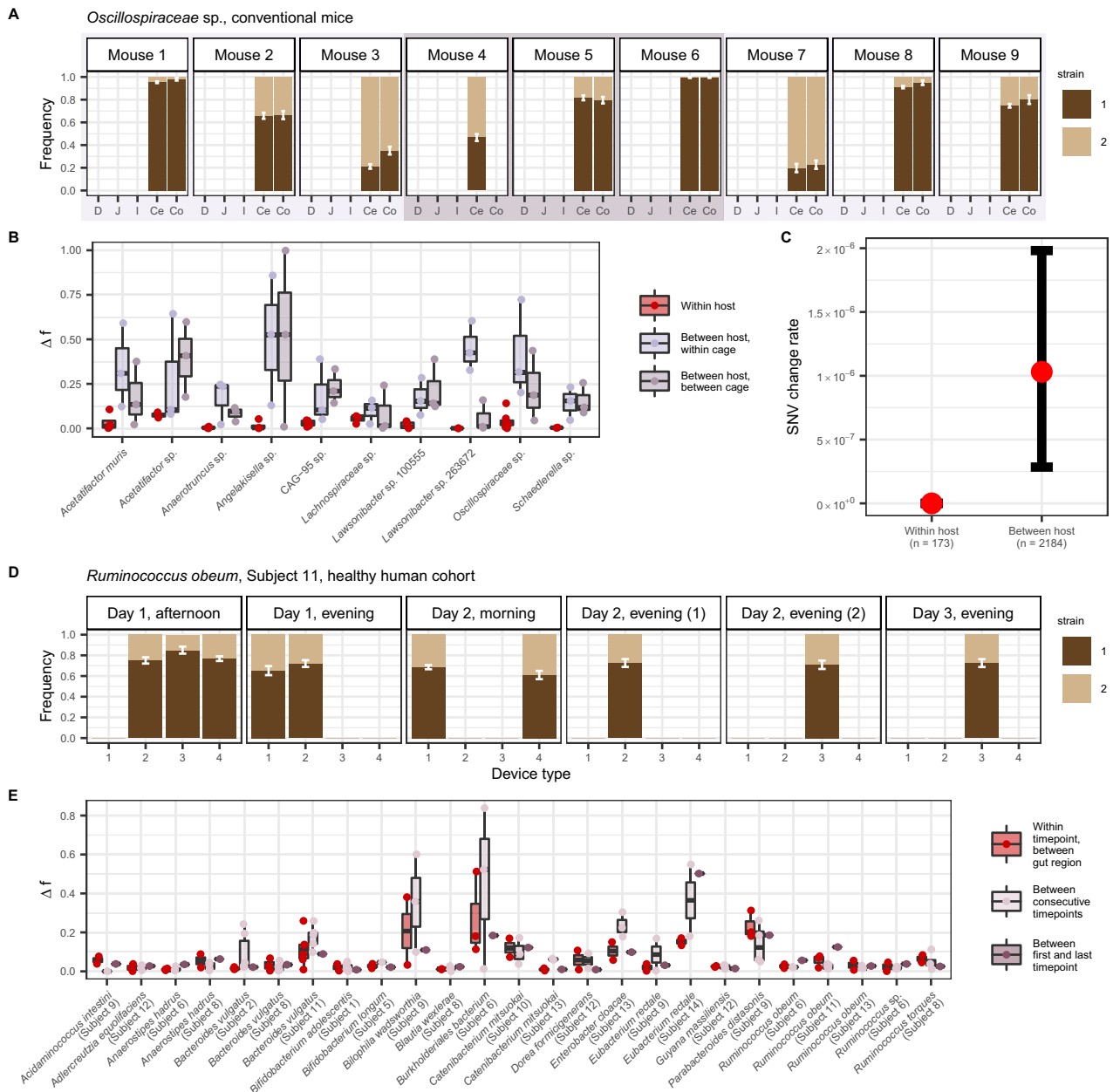

inoculum (minimum of 20 reads), 15 had non-zero frequency in the inoculum, indicating that SGV is an important source of genetic variation in these mice (Fig. 5A, Supplementary Fig. 8, and Supplementary Data 8). Of these, five nonsynonymous and one synonymous SNVs were observed to undergo parallel sweeps from low frequency in the inoculum to high frequency in multiple independent cages. One of these parallel SNVs was in *P. distasonis* in a gene encoding a TonB-dependent transporter, which rose from low frequency in the inoculum ($f = 0.057$) to high frequency in multiple mice in all three cages (ranging from an average $f = 0.319$ in mouse 4 to an average $f = 1.00$ in mouse 6), consistent with parallelism (Fig. 5A and Supplementary Data 8). TonB-dependent transporters are ubiquitous outer membrane-associated proteins involved in nutrient transport[43] with structural variation having been found to be associated with niche partitioning by different *Bacteroides* taxa[44]. Moreover, TonB-dependent transporters have been found to be a common target of adaptation in the human gut microbiome[41,45,46]. Notably, all the SNVs observed to undergo parallelism were not found to be differentiated along the gut within hosts. This fact would seem to suggest that even

when SNVs are adaptive, they are either broadly adaptive to the entire gut, or alternatively have high migration rates that outweighs local selection pressures, perhaps due to mixing and peristalsis along the gut[47].

## Genetic homogeneity is generalizable to the gut of conventional mice and healthy humans

Having established that uniform strain frequencies and spread of evolutionary adaptations along the gut is a common feature in humanized mice, we next investigated whether these same patterns also arise in hosts with established native microbiota, specifically conventional mice and healthy humans.

We first analyzed a cohort of nine conventional mice that were cohoused until weaning at three weeks of age before being housed into three cages of three mice each. At six weeks of age, we sampled and sequenced luminal contents from the same five gut regions that were previously examined in humanized mouse guts ("Methods"). Large intestinal samples harbored higher alpha diversity and a different family-level composition than small intestinal samples ("Methods")

**Fig. 6 | Uniform genetic variation along the gut in conventional mice and healthy humans. A** Strain frequencies for two co-colonizing strains of an *Oscillospiriceae* species long the guts of nine conventional mice housed across three cages. "Strain 1" and "strain 2" are denoted by dark and light brown bars, respectively. The strain frequency in each sample is calculated as the mean of the frequency of SNVs belonging to that strain cluster, and error bars represent bootstrapped 95% confidence intervals. Mice 1–3 were co-housed in cage 1, mice 4–6 in cage 2, and mice 7–9 in cage 3, as indicated by the light and dark purple boxes. Absence of a strain frequency bar indicates insufficient coverage or species absence in a given sample. See Supplementary Fig. 10 for additional strain frequency plots corresponding to species with co-colonizing strains in the conventional mice and Supplementary Data 9 for precise strain frequency values. **B** Change in strain frequency $\Delta f$ was calculated for ten species present in multiple mice, cages, and gut regions ("Methods"). Each dot represents the change in major strain frequency between a pair of samples from within the same host (red), between hosts in the same cage (light purple), and between hosts in the different cages (dark purple). The pairwise comparisons plotted here have been chosen to reduce redundancy (e.g., only a single within-host $\Delta f$ value is plotted per species per mouse) ("Methods"). Boxplot center lines represent the median, hinges represent the first and third quartiles, and the whiskers extend to the maximum and minimum point of each distribution. See Supplementary Data 10 for precise $\Delta f$ values for all pairs plotted. **C** Comparison of the rate of SNV frequency changes within and between mice. Colored in red are the observed SNV frequency change rates aggregated across species. Bars represent bootstrapped 95% confidence intervals constructed using the 2.5th and 97.5th percentiles of 1000 bootstrap iterations, with the values in each iteration representing the overall SNV allele frequency change rate aggregated across species and sample pairs after randomly sampling 100 quasi-phaseable genome pairs with replacement ("Methods"). The total number of quasi-phaseable genome pairs used to calculate the point estimates

and bootstrapped confidence intervals of allele frequency change rate are shown below the *x*-axis label for each category. The allele frequencies of SNVs experiencing an extreme allele frequency change between pair of mouse samples can be found in Supplementary Data 11. **D** Strain frequencies for two co-colonizing strains of *Ruminococcus obeum* were inferred from capsule devices that sampled luminal contents along the gut of a healthy human subject (subject 11) across six timepoints. Strain frequencies, colors, and error bars are defined as in panel A. Device type 1 targeted the pyloric sphincter to the upper small intestine, device type 2 targeted the upper to mid-small intestine, device type 3 targeted the mid- to lower small intestine, and device type 4 targeted the lower small intestine into the ascending colon[48]. Timepoints represent the time at which capsules were swallowed, with days being relative to the first timepoint plotted and times of day being coarsened into "morning" (before 12 pm PST), "afternoon" (after 12 pm and before 8 pm PST), and "evening" (after 8 pm PST) bins. See Supplementary Fig. 12 for additional strain frequency plots corresponding to species x host pairs with co-colonizing strains and Supplementary Data 12 for precise strain frequency values. **E** Change in strain frequency $\Delta f$ was calculated between samples belonging to 25 species x host pairs with co-colonizing strains present in multiple timepoints and device types ("Methods"). Boxplot center lines, hinges, and the whiskers extend are the same as in (**B**). Each dot represents the change in major strain frequency between samples from different device types swallowed during the same timepoint (red), collected during consecutive timepoints (light purple), and collected during the first and last timepoint (dark purple). The pairwise comparisons plotted here have been chosen to reduce redundancy (e.g., only one device type pair is used per within-timepoint sample pairs, representing the farthest possible pair of measurements along the gut) ("Methods"). See Supplementary Data 13 for precise $\Delta f$ values for all pairs plotted. Strain frequency change over the course of the entire sampling period rarely exceeded changes occurring between consecutive timepoints, suggesting that hourly or daily shifts in strain frequency are transient.

(Supplementary Fig. 9), confirming that the expected spatial organization does exist at higher taxonomic levels along the guts of the conventional mice.

Further replicating our findings from humanized mice, we found that in conventional mice, co-colonizing strains of the same species exhibited largely uniform frequencies across gut regions. This pattern contrasts with the substantial variation in strain frequencies between mice, even if co-housed (Fig. 6A, Supplementary Fig. 10, and Supplementary Data 9). Among the ten bacterial species with co-colonizing strains that were present in multiple mice, cages, and gut regions, median change in frequency $\Delta f$ between gut segments within hosts was only $2.21 \times 10^{-2}$, an order of magnitude smaller than the median frequency changes observed between mice in the same cage ($2.16 \times 10^{-1}$) and in different cages ($1.42 \times 10^{-1}$) (Fig. 6B and Supplementary Data 10). Corroborating these results, gut region explained the least variation (average variance explained = 0.75%), whereas mouse identity explained the most variation (average variance explained = 78.39%) (Supplementary Fig. 11). Cage explained a moderate amount of variation (average variance explained = 19.97%). These results indicate that uniform frequencies of strains along the gut is not unique to humanized mice and may be a common feature of mouse microbiota.

Additionally, we did not observe evolutionary changes accruing between gut regions within hosts even while changes arose between hosts at a rate of $1.03 \times 10^{-6}$ changes/bp (Fig. 6C and Supplementary Data 11). These results indicate that any evolutionary changes arising within hosts sweep globally throughout the gut.

Next, to determine whether spatially uniform patterns of genetic diversity are a feature of human guts as well, we analyzed previously published metagenomic data[48] collected using ingestible sampling capsules designed to sample luminal contents from four successive regions of the gastrointestinal tract: pyloric sphincter to the upper small intestine, upper to mid-small intestine, mid- to lower small intestine and the lower small intestine into the ascending colon. In this dataset, 15 healthy adult subjects ingested four capsule types twice daily for as long as a week.

We hypothesized that strain frequencies in humans might vary more along the gut than in mice, given the greater length and compartmentalization of the human intestine, as well as the potential for host-specific niche adaptation over long residence times[25,27]. Contrary to this expectation, strain frequencies were generally homogeneous not only along the gut but also across timepoints. For example, the frequency of the major strain of *Ruminococcus obeum* in Subject 11 fluctuated around a mean of 72% with a variance of 0.4% across gut locations and timepoints (Fig. 6D and Supplementary Data 12). More generally, across the 25 species x host pairs with co-colonizing strains in multiple timepoints and device types, the median change in major strain frequency $\Delta f$ between gut regions within the same timepoint was only $3.79 \times 10^{-2}$. This was the same order of magnitude as the median change observed between consecutive timepoints ($2.92 \times 10^{-2}$), or even between the first and last observed timepoint for a species x host pair ($3.55 \times 10^{-2}$) (Fig. 6E and Supplementary Data 13). These results indicate that strain frequency tends to be uniform along the human gut as well as across hourly and daily timescales.

While the majority of species x host pairs never showed a temporal or spatial strain fluctuation in strain frequency exceeding 25%, there were several outlier species x host that harbored strains undergoing large frequency temporal shifts, and to a lesser degree, spatial shifts. However, these fluctuations tended to be transient. For example, the frequency of the major strain of *Burkholderiales bacterium* in Subject 6 fluctuated between a minimum of ~10% and maximum of ~95% within and across timepoints, but displayed no consistency in the direction of change across either space or time (Supplementary Fig. 12). We observed this in other species x host pairs as well, illustrating that strain fluctuations that do occur along the gut and over time are short-lived, consistent with previous work showing that stably co-colonizing strains can fluctuate in frequency around a carrying capacity over longer timescales[25].

We next asked whether extreme SNV frequency changes arising as a consequence of within-host evolution were spatially segregated. To do so, we identified SNV frequency changes between distinct regions of the gut for each of the 56 species x host pairs (representing

35 species detected in 11 subjects) with enough data in at least two capsules for a given time point. Out of the 56 pairs, only 7 of these (12.5% of species $x$ host pairs) had detectable SNV frequency changes between gut regions within a single timepoint (Supplementary Data 14), indicating that spatial segregation of SNVs is a feature in only a minority of samples.

Finally, we investigated whether the SNV changes arising along the gut are observable over multiple time points or whether they are transient. Persistence would suggest that environmental gradients maintain spatial segregation of genetic variation along the gut over time. Of the seven species $x$ host pairs with spatial differences, four had sufficient temporal and spatial data (at least two timepoints with at least two device types per timepoint). In three of four species $x$ host pairs, spatial differences did not persist over multiple time points, with allele frequencies often reverting to their original state (Supplementary Fig. 13 and Supplementary Data 14). These results imply that sustained extreme allele frequency changes along the gut are rare.

## Discussion

In this study, we analyzed the frequencies of bacterial strains and evolutionary changes along the guts of eight mice inoculated with the same human-stool-derived inoculum. In sharp contrast to the species-level heterogeneity reported here and in previous studies[3,6,28,29,49–57], we found substantial *genetic* homogeneity along the gut (Figs. 4 and 5 and Supplementary Figs. 5 and 8). Spatial uniformity was a feature of not only the bacterial species analyzed in humanized mice, but also those in conventional mice and healthy humans, indicating that uniformity is a robust feature of mammalian gut ecosystems. Together, these results indicate that spatial segregation of strains is not a requisite for their coexistence and that environmental gradients do not necessarily generate large strain fluctuations or allele frequency differences along the gut lumen.

Uniformity likely arises due to rapid migration rates of bacteria between gut regions and, in the case of the mice studies, due to rapid migration between mice (Figs. 4–6 and Supplementary Figs. 5–7, and 11). This latter conclusion is supported by the observation that strain frequencies are more similar between co-housed mice than between mice housed in different cages, consistent with recent work demonstrating extensive social transmission of bacteria among co-housed mice[14,35]. Additionally, this conclusion is supported by the observation that evolutionary changes at the SNV level are usually found at similar frequencies along the gut, suggesting migration of variants arising from a single origin as opposed to the same evolutionary change arising in different locations along the gut independently of each other. The possibility of rapid migration facilitating uniformity is consistent with recent work showing that there is significant mixing of luminal contents along the gut[47].

Uniform strain frequencies are not only found along the length of the gut but also over time, reflecting stable coexistence of strains. Stable coexistence is a common feature of healthy human gut microbiomes[25,27] and in experimental evolution studies[58,59]. Our own analysis of the healthy human cohort sampled over multiple time points show that while strains tended to fluctuate in frequency more across time than they did across space (i.e., between capsule) within any one timepoint, these temporal strain frequency fluctuations were short-lived and tended to revert to the overall trend at subsequent sampling points (Fig. 6E and Supplementary Fig. 12). Stable coexistence of conspecific strains is thought to be facilitated at least in part by metabolic niche partitioning[60]. Our findings support the role of metabolic niche partitioning in stably maintaining strain diversity, as spatial segregation does not seem required to reduce competition among strains.

Future work employing higher-resolution sampling strategy across both time and space could reveal bacterial genetic heterogeneity that was not observed in this study. For example, sampling the gut prior to eight weeks post-inoculation might have captured distinct spatial distribution of alleles before they migrated to all gut regions. Additionally, in this study, we focused on extreme allele frequency changes, as these are unlikely to be confounded with sampling error in shotgun data and instead are likely to have arisen only due to adaptation. However, more subtle allele frequency differences may manifest along the gut, especially among neutral genetic variants, consistent with recent human microbiome studies analyzing metagenome-assembled genomes reporting locally restricted, potentially neutral evolutionary changes in a limited number of species[16,17].

In addition, while we focused on luminal contents from five gut regions and observed uniform genetic variation, other physiological structures found in the mammalian gut could exhibit greater spatial heterogeneity. For example, different crypts in the intestinal epithelium could potentially be occupied by distinct strains, similar to that of skin pores[61]. Crypts are known to have communities distinct from the lumen[49,62], and may perform ecological filtering at the strain and nucleotide-level as well. Similarly, the mucosa displays different biogeography from the lumen[50], and mucosal surfaces are known to generate population structure in culture[63], promote site-specific binding in certain bacterial species that have co-evolved with their hosts[64], and drive adaptive differentiation relative to populations in the gut lumen[9]. However, the technical challenges in obtaining sufficient bacterial biomass for metagenomics sequencing from crypts or mucosa is significant[65]. The continued improvement in host-depletion and fine-scale sampling strategies (e.g., Wu-Woods et al.)[17] will enable investigation of bacterial population structure at unprecedented spatial resolution.

Our findings in healthy mammalian hosts raise the question of whether spatial genetic structure may emerge under disease conditions or altered environments. Local inflammation or environmental changes could generate distinct niches that foster local bacterial adaptation or filter for specific, disease-adapted strains. In future work, it will be interesting to investigate if spatial structure at the genetic level arises in diseased hosts. Intestinal disorders such as inflammatory bowel disease are known to generate local inflammation phenotypes and disrupt gut microbiome composition[66,67] in specific gut regions[4,68,69]. For example, in a recent mouse study, adaptive SNVs in a species of pathogenic bacteria were found to appear locally in different colonic tissues, enabling translocation of a pathogenic bacteria from the colonic lumen to the mucosa, and subsequently to the liver[9]. Thus, in pathogenic bacteria and disease scenarios, we might observe specific strains or local adaptations associated with these acute conditions along the gut, supporting our original hypothesis that environmental heterogeneity might induce genetic diversity along the gut.

In conclusion, our findings demonstrate that bacterial strains and their evolutionary changes are uniformly distributed along the gut lumen of across different mammalian hosts, in contrast to the spatial heterogeneity observed at higher bacterial taxonomic levels. These findings indicate that spatial segregation of strains is not a requisite for their co-existence, and that environmental gradients do not result in local adaptation among alleles experiencing the largest allele frequency changes. Future studies aimed at understanding the dynamics of migration facilitating this uniformity and the metabolic strategies employed by strains to co-exist will be crucial for elucidating the dynamics of strain-level diversity and its functional consequences within the gut ecosystem.

## Methods

### Experimental design

All animal experiments were performed in accordance with the University of British Columbia Animal Care Committee (protocols A19-0078 and A23-0115) and in direct accordance with the guidelines of the Canadian Council of Animal Care. Swiss-Webster mice, bred in-house, were housed in Ehret cages for the duration of the study, in a 12/12 light

cycle, with temperature between 68 °F and 79 °F and humidity between 30% and 70%. All mouse experiments and breeding were conducted in the Centre for Disease Modelling rodent facility at the University of British Columbia, Canada. Human samples were collected in accordance with the University of British Columbia Office of Research Ethics (protocol H21-02464). Human subjects provided written informed consent for use of their samples and data in this study.

For the humanized mouse study, we performed shotgun sequencing on the luminal contents of mice we previously analyzed with 16S rRNA sequencing[3]. Swiss-Webster male and female mice at nine weeks of age were used and were provided with an autoclaved standard diet (Purina LabDiet 5K67). Briefly, eight germ-free Swiss-Webster mice were orally gavaged with the same human stool sample (the inoculum) from a healthy adult human previously labeled TL1[3]. This set up is akin to previous studies conducting controlled mouse studies with human inoculums[18–21]. The eight mice were housed in three different cages according to sex and assigned treatment in the following configurations: cage 1 had mice 1–3 (sex male), cage 2 had mice 4 and 5 (sex male), and cage 3 had mice 6–8 (sex female). Given that human microbiota can rapidly colonize the mouse in a matter of days to weeks, we allowed for a total of eight weeks to colonize and equilibrate in the mice[18–21], allowing for stabilization of community structure and recovery following oral gavage. This period included six weeks on a standard rodent diet (LabDiet 5k67), followed by two weeks in which mice in cages 2 and 3 were exposed to a fiber-rich diet of 30% guar gum (TestDiet 5BSE) while those in cage 1 remained on the same chow. With this experimental set up, we were able to ascertain replicability and generalizability of the genetic homogeneity observed along the gut in three independent cages with varying diets and sexes, as well as contrast within-host, across-host, and across-cage dynamics (Fig. 1).

Eight weeks after inoculation, mice were sacrificed using carbon dioxide with secondary cervical dislocation, and luminal contents were collected from five intestinal regions: the duodenum, jejunum, ileum, cecum, and colon. The intestines were carefully sectioned using a razor blade into the corresponding regions and the contents were squeezed into 1.5 mL microcentrifuge tubes. DNA extraction was performed on luminal content samples and the original inoculum (TL1) from 96-well plates using the DNeasy PowerSoil Pro Kit (Catalog number: 47016). For more information on the approach used, see the "Methods" section of Ng et al.[3].

Conventional mice microbiomes were obtained from nine male Swiss-Webster mice from the same litter and with their own microbiomes at six weeks of age. Mice were cohoused until three weeks of age, at which point they were separated into three cages (three mice per cage). They were provided with an autoclaved standard diet (Purina LabDiet 5K67) for the duration of the six weeks. At six weeks, mice were sacrificed, and luminal contents were collected from the same five intestinal regions as with the humanized mice (duodenum, jejunum, ileum, cecum, and colon), and DNA extraction was performed using an identical approach.

## Metagenomic sequencing

Metagenomic sequencing was performed on luminal content samples from humanized and conventional mouse cohorts and the inoculum using a NovaSeq with 150 bp paired-end reads. Humanized mouse samples were sequenced to a depth ranging from 23 M to 136 M reads (median = 33 M reads), and the inoculum was sequenced to a depth of 119 M reads. Samples from mice 7, 8, and the inoculum were sequenced in a separate batch than from samples from mice 1–6 at the highest end of the range of sequencing depths. Conventional mouse samples were sequenced to a depth ranging from 39 to 75 M reads (median = 45 M reads) (Supplementary Data 1).

## Estimation of species abundances, SNVs, and CNVs

For the humanized mice and data from Shalon et al.[48], we used the Metagenomic Intra-species Diversity Analysis System 1 (MIDAS 1; version 1.2, downloaded November 24, 2021)[26] to calculate relative species abundances, gene copy number variants (CNVs), and SNVs. MIDAS 1 calculated these values by mapping reads against a database of genomes representing 5952 bacterial species[26,70] found in the human gut microbiome.

The conventional mice microbiomes harbored bacteria native to mice and thus was not well-suited to the MIDAS 1 database used to process the humanized mouse and human microbiomes, as the reference genomes in the database corresponded only to human-associated bacterial species, resulting in fewer reads being mapped. We therefore used MIDAS 3 (version 3.0.1; downloaded August 5, 2024) to construct a mouse-specific database using isolate and metagenome assembled genome reference genomes from the Mouse Gastrointestinal Bacteria Catalogue[71] using a standard approach[72]. MIDAS 3 was subsequently used to map reads to this database and calculate relative species abundances, CNVs, and SNVs.

While the outputs from both pipelines were generated with permissive parameters, as described below, we subsequently applied several stringent post-processing filters to rule out metagenomic artifacts and to accurately estimate species, CNVs, and SNVs. The post-processing pipeline we use is also briefly described below and reflects the pipeline developed in Garud et al.[23].

## Estimating species relative abundance

For humanized mouse and Shalon et al., 2023 data, MIDAS 1 was used to estimate species presence and abundance by mapping reads to 15 single-copy marker genes from 5952 representative species[26,70]. Reads were mapped with default MIDAS 1 settings. Species with a median single copy marker gene coverage ≥3 reads were marked as present within a sample. These reavd coverages were then used by MIDAS 1 to compute species relative abundance. For conventional mouse data, MIDAS 3 was used estimated species presence and abundance in a similar manner, this time to 15 single-copy marker genes from 1093 representative species[71] in the custom-built mouse-specific database. Species were marked as present if they had a median marker coverage ≥2 reads, per the recommendation of MIDAS 3 developers[72,73].

## Quantifying pathway relative abundance

To quantify changes in pathway abundances along the gut, we used HUMAnN 3 (Beghini et al., 2021) version 3.9. HUMAnN3 was downloaded on August 11, 2025 and run with default settings using the CHOCOPhlAn pangenome database version v201901_v31 and the UniRef90-DIAMOND 2019-01 protein database obtained through the HUMAnN3 download utility both on August 11, 2025. UniRef gene families were mapped to KEGG Ortholog (KO) groups using the HUMAnN3-provided mapping file map_ko_uniref90.txt.gz (downloaded December 5, 2025). Gene family and pathway read counts were summed across all taxa to generate community-wide functional profiles. Pathway abundance was automatically normalized in the MaAsLin3 analysis described below via the "total sum scaling" setting.

To identify pathways differing between the upper and lower gut, we analyzed the raw pathway count matrix using the MaAsLin3[33] R package version 1.2.0. MaAsLin3 was downloaded on December 7, 2025 and run with the default parameters on the raw pathway count matrix (minimum feature abundance of 0.0001, minimum prevalence of 0.1, and default FDR correction). MaAsLin3 fits separate multivariate models per pathway. Abundance associations were assessed using linear regression applied to $\log_2$-transformed, total-sum-scaled relative abundances in samples with nonzero feature counts; $p$-values reflect a two-sided $t$-test of each coefficient against the median coefficient across features (median comparison) to account for compositionality. All models included intestinal position, cage, and read counts

as fixed-effect covariates stratified by subject ID. Reported *q*-values are false discovery rate (FDR)-

adjusted *p*-values computed using the Benjamini–Hochberg method. Significance threshold: *q* < 0.05. *P*-values were adjusted for multiple comparisons using the Benjamini–Hochberg false discovery rate (FDR) procedure, as implemented in MaAsLin3, and pathways with *q* < 0.05 were considered significant.

## Quantifying SNV frequencies

We estimated SNV frequencies for each species in each sample. Downstream, these frequencies were used to estimate nucleotide diversity, identify quasi-phaseable samples, and quantify SNV frequency differences between pairs of quasi-phaseable samples. To estimate SNV frequencies in humanized mouse and Shalon et al., 2023 samples, reads were mapped to representative reference genomes using default MIDAS 1 mapping thresholds and arguments[23]. SNV frequencies were estimated in conventional mouse data using MIDAS 3, with mapping thresholds and arguments set to match MIDAS 1 defaults.

To obtain accurate estimates of SNV frequencies, we imposed coverage requirements. MIDAS 1 and 3 report the read coverage *D* for each site in the reference genome for each sample. We excluded any sample with a mean *D* < 5× across all protein coding sites, as was done in Garud et al., 2019. We further excluded from analysis species nonzero coverage at ≥ 40% of reference sites within a sample. Additionally, to minimize the possibility of any mapping errors, we implemented two controls. First, as described below and as was done in Garud et al.[23], we exclude genes with copy number greater than 3 in at least one sample in our study, as they were potential candidates for read stealing. Second, we exclude individual sites with a coverage value < $0.3\bar{D}$ or > $3\bar{D}$ where $\bar{D}$ is mean genome-wide coverage, as these sites had abnormal coverages outside the range of expectations for single copy genes.

Finally, SNV frequency changes were identified only if median coverage values between two samples from the same mouse were within a factor of three of each other, as we expect that coverage should be relatively constant within a mouse. Below, we describe additional coverage filters that were imposed for various calculations, including SNV frequency changes.

## Annotating SNVs as synonymous versus nonsynonymous

MIDAS 1 annotated SNVs as synonymous and nonsynonymous based on the reading frames of genes as annotated in the PATRIC database[74], and MIDAS 3 annotated SNVs as synonymous and nonsynonymous based on annotations from Prokka[75].

## Quantifying gene copy number

We estimate copy number variation (CNV) to exclude any genes from further analysis that may be potential candidates for read stealing or donating. To estimate CNVs in humanized mice and Shalon et al., 2023 data, reads were mapped to pangenomes using default MIDAS 1 mapping thresholds and arguments[23]. CNVs were estimated in conventional mouse data using MIDAS 3 with mapping thresholds and arguments set to match MIDAS 1 defaults.

To minimize read mapping errors, we identified genes with abnormally high gene copy number values, which may be potentially indicative of erroneous read recruitment[23]. MIDAS 1 and 3 estimates CNVs by mapping reads to a *pangenome* consisting of several sequenced isolates belonging to the same species. Average gene coverage was calculated as the number of reads mapping to that gene normalized by gene length. Both versions of MIDAS compute copy number, *c*, as the ratio between a gene's average coverage and the median coverage of a set of 15 single-copy marker genes belong to that species. Genes with a copy number *c* > 3 in at least one sample in our study were excluded during post-processing, as they were potential

candidates for stealing. To exclude any additional genes that may suffer from erroneous read recruitment, we further filtered out any genes with a copy number *c* > 3 in the human samples analyzed in Garud et al.[23]. Finally, any genes sharing ≥ 95% average nucleotide identity with any other gene across species boundaries as identified in a set of "blacklisted" genes in Garud et al., 2019 (section A.iv)[23] were excluded to further prevent erroneous read mapping.

## Calculating alpha and beta diversity of gut samples

Alpha diversity was computed for humanized and conventional mouse data using the Shannon diversity index as implemented by the Vegan package in R[76] (Fig. 2A and Supplementary Fig. 9A). To assess if Shannon diversity between gut regions was significantly different, we used a paired Wilcoxon signed-rank test comparing pairs of samples from one region versus the other from the same mouse. To calculate fold change in alpha diversity between the small and large intestines, alpha diversity was averaged across small intestinal regions (duodenum, jejunum, and ileum) and large intestinal regions (cecum and colon), and the per mouse fold change was calculated as the ratio of the averaged large intestinal and small intestinal alpha diversity values. The fold change values presented in the main text represent the average of these per-mouse estimates.

To produce the stacked bar plots in Fig. 2B and Supplementary Fig. 9B, species abundances within taxonomic families were summed. Species with less than 0.1% abundance were excluded. To test for differential abundance of specific families between the small and large intestines of humanized mice, relative abundances for each family were averaged across the small intestine (duodenum, jejunum, and ileum) and large intestine (cecum and colon) in each host, the wilcox.test() in R was used to apply a paired Wilcoxon signed-rank test to compare average large and small intestinal relative abundances (paired by mouse) for each species, generating per-species estimates for the median log2 fold change in mean relative abundance between large and small intestinal samples in the same mouse, as well as *p*-values and 95% confidence intervals to assess significance (Supplementary Fig. 1).

## Calculating nucleotide diversity

For every humanized mouse and sample and the inoculum, we estimated *π*, a population-level metric of nucleotide diversity that represents the probability of two randomly sampled genomes from the population have different alleles at a base pair. To do so, we used the formula for nucleotide diversity applied in Schloissnig et al.[77], which accounts for total read counts:

$$\pi(S, G) = \frac{1}{|G|} \sum_{i=1}^{|G|} \sum_{B_1 \in ATGC} \sum_{B_2 \in ATGCB_1} \frac{x_{i,B_1}}{D_i} \frac{x_{i,B_2}}{D_i - 1} \qquad (1)$$

Here, *S* is the sample; *G* is the genome of the focal species, with $|G|$ being its size; *i* is the locus in the genome; $B_1$ and $B_2$ are the reference and alternative alleles, respectively; $x_{i,B_j}$ is the number of reads with allele $B_j$ at locus *i*; and $D_i$ is the total read depth at locus *i*. This formula is an extension of a previously proposed *π* estimator on next-generation sequencing data devised by Begun et al.[78]. Sites with read coverage less than four were excluded from *π* calculations, such that $|G|$ is equivalent to the total number of sites with adequate coverage rather than the total size of the genome.

Samples from mice 7 and 8 were sequenced to a substantially higher depth compared to samples from mice 1 through 6 (Supplementary Data 1), resulting in elevated levels of coverage per bp for bacterial species in mice 7 and 8. As a consequence, there was an increased opportunity to detect higher levels of nucleotide diversity in mice 7 and 8 merely due to systematic differences in coverage. To be able to compare nucleotide diversity levels between mice, we applied a

random down-sampling per nucleotide site such that the depth of mice 7 and 8 matched the median coverage of mice 1 through 6.

Distributions of $\pi$ in Fig. 3A were plotted for the 30 species which met the following criteria: minimum coverage of four reads in the inoculum and at least three mouse samples representing at least two different mouse hosts. Species were only considered in samples in which at least 500,000 nucleotides met the minimum coverage threshold.

## Calculating $F_{ST}$

$\pi$ estimates were used to compute genome-wide $F_{ST}$ as a quotient of the mean genome-wide $\pi$ subtracted from 1:

$$F_{ST}(S_1, S_2, G) = 1 - \frac{\pi_{within}}{\pi_{between}} = 1 - \frac{(\pi(S_1, G) + \pi(S_2, G))/2}{\pi(S_1, S_2, G)} \quad (2)$$

Where $\pi_{within}$ is given by the formula 1, and pairwise $\pi_{between}$ is given by the following formula also derived in Schloissnig et al., 2013:

$$\pi(S_1, S_2, G) = \frac{1}{|G|} \sum_{i=1}^{|G|} \sum_{B_1 \in ATGC} \sum_{B_2 \in ATGC \setminus B_1} \frac{x_{i,B_1,S_1}}{D_{i,S_1}} \frac{x_{i,B_2,S_2}}{D_{i,S_2}} \quad (3)$$

Where $k$ in $S_k$ designates sample 1 or 2, respectively. We used ileum and colon to represent the small and large intestine, respectively, because those regions had the highest mean read coverage across all samples in each gut region. In order to accurately estimate allele frequencies, each site was only included in the $F_{ST}$ calculation if a minimum of 4 reads was observed at that site for both samples being compared.

## Inferring strain frequencies

Previous work has shown that hosts tend to be colonized by multiple genetically distinct strains (typically 1–4) of the same species[23,79]. When levels of recombination between strains within a host is sufficiently low, these strains may co-exist as genetically distinct subpopulations[24]. Typically, two randomly drawn strains from different hosts harbor on order $10^3 \cdot 10^4$ SNV changes[23,41,77,80], and two strains residing within a host are expected to be similarly diverged[23]. We leverage this expected divergence between strains to distinguish between the strain genotypes and infer their overall frequencies.

Inferring strain genotypes and frequencies from short read sequencing data is a difficult problem because linkage between cosegregating SNVs residing on the same strain backbone are destroyed by the fact that they are unlikely to reside on the same short read. Even perfectly linked SNVs may be difficult to assign to the correct strain given the potential for sampling noise. However, recent work has shown that strain inference is feasible when there are many samples collected from the same hosts, as alleles belonging to the same strain will display highly correlated allele frequencies across samples[24,25,59,81–83].

We leverage recent work by Roodgar et al.[24] and extended by Wolff et al.[25] to infer strain identity and frequency in samples from all three datasets analyzed in this study. The papers cited above inferred strain genotypes and frequencies from *temporal* data. Here, we use *spatial* data both within mice and across mice, and a mixture of *spatial* and *temporal* data within human hosts sampled in Shalon et al., 2023 to infer strain genotypes and frequencies within samples. Because humanized mice were inoculated with the same human sample and conventional mice shared homogenized microbiomes due to cohousing, we inferred strain frequency jointly across all mouse samples belonging to the same dataset, reasoning that mice from the same dataset all shared the same subset of strains. For Shalon et al., 2023 data, we inferred strain frequency jointly across all samples belonging to the same host, likewise reasoning that these samples carried the same subset of strains.

Briefly, the approach clusters SNVs segregating on the same strain background by detecting correlations between SNVs with highly similar allele frequencies across samples. The degree of correlation between a pair of SNVs $i$ and $j$ is measured by distance metric $d$ between allele frequency trajectories $\hat{f}_i$ and $\hat{f}_j$:

$$d(\hat{f}_i, \hat{f}_j) = \frac{1}{S} \sum_{s=1}^{S} \frac{2(D_{is} + D_{js})(\hat{f}_{is} - \hat{f}_{js})^2}{(\hat{f}_{is} + \hat{f}_{js})(1 - \hat{f}_{is} + 1 - \hat{f}_{js})} \quad (4)$$

where $S$ is the number of samples, $\hat{f}_{is}$ and $\hat{f}_{js}$ are the frequency of the alleles in a sample $s$, and $D_{is}$ and $D_{js}$ represent the read depth of the alleles in sample $s$. SNVs that are in perfect linkage with one another are expected to have a $d = 0$.

To ensure that high-quality and informative sites were included in our stain inference procedure, we only considered loci that had a read coverage of $D \geq 10$ and were polymorphic in at least 20% of samples in which the species was detected. Filtering out non-polymorphic loci prevented the algorithm from generating clusters at allele frequencies of 1 or 0 that reflect low frequency variation within a strain population as opposed to actual genetically distinct strains.

Detecting correlated clusters of SNVs belonging to the same strain background can be confounded without knowing if the alternative or reference allele belongs to one strain versus another. To address this issue, as done in Roodgar et al.[24], we set $d$ between a pair of loci to the minimum of $d(\hat{f}_i, \hat{f}_j)$ and $d(\hat{f}_i, 1 - \hat{f}_j)$ to ensure that polarization differences (i.e., calling an allele frequency 0.2 instead of 0.8, based on the reference allele used in the reference genome) between loci did not prevent them from being clustered together.

To identify SNVs linked on the same strain background, we applied a greedy, network-based algorithm to extract large cluster SNVs with highly correlated allele frequencies across samples. The algorithm begins by forming a network in which nodes represent SNVs, with pairs SNVs connected by an edge if they have a distance $d < 3.5$, which represents a maximum distance threshold below which pairs of SNVs are likely to be linked on the same strain background, as demonstrated in Roodgar et al.[24]. Next, the algorithm identifies a focal SNV as the SNV with the maximum number of connections (i.e., edges) and extracts all polymorphic SNVs connected to it. This cluster of SNVs was designated as a strain if it included at least $10^3$ SNVs, as this represents the lower bound for the typical number of alleles that are expected to segregate between genetically distinct strains[23,41,77,80]. Note that SNVs connected to the focal SNV were only extracted if they were also connected to at least 25% of other SNVs within a cluster. SNVs included in the initial cluster were removed from the overall set of variable sites, and the clustering process was repeated until no additional clusters of at least $10^3$ SNVs were identified. Strain frequencies were then inferred as the mean SNV allele frequencies in each cluster (Supplementary Fig. 14). However, despite requiring strain clusters to have a minimum of $10^3$ SNVs, not all SNVs met our coverage requirement of $D \geq 10$ across all samples. Therefore, we only report strain frequencies for samples in which every cluster has at least 100 high coverage SNVs ($D \geq 10$) to ensure that the inferred strain frequencies were well-supported.

95% confidence intervals for strain frequencies in each sample were generated using a stationary bootstrapping approach[84]. Specifically, 1000 bootstrap samples, each consisting of 100 SNVs were drawn with replacement from each strain cluster within each sample, and subsequently, the mean of the subsampled SNV frequencies in each bootstrap sample was computed as strain frequency. The upper and lower bounds of the 95% confidence interval was calculated as the 2.5 and 97.5 percentile of the bootstrapped strain frequency distribution.

In addition to the filters used by Wolff et al., 2023, we required minor alleles to have a read support of at least four reads to prevent

rare mutations and sequencing errors from erroneously giving rise to spurious low frequency clusters inferred from conventional and humanize mouse data. For strain frequency profiles in Shalon et al., 2023, we filtered out spurious low frequency clusters and lowly abundant strains using a different approach by requiring all strains rise above a frequency of 5% in at least one sample to be included in downstream analyses.

We visually inspected all strain frequency profiles and either applied manual modifications to the algorithm or excluded from downstream analysis all profiles which violated one of the following expectations. In a scenario when only one strain is present, we expect *zero* clusters to be detected. In a scenario in which two strains are present, we expect only *one* cluster, representing the alleles segregating between the two strains. The relative frequencies of the two strains can be inferred as $\bar{f}$ and $1 - \bar{f}$, where $\bar{f}$ is the mean frequency of the clustered SNVs. In a scenario in which three strains are present, we expect *three* clusters, the frequencies of which roughly sum to 1 in each sample, representing the SNV segregating between each of the three pairs of strains. This second scenario did not occur in our humanized or conventional mouse data. In the Shalon et al., 2023 data, the strain frequency inference pipeline did at times yield a maximum of three strain clusters. However, manual inspection revealed that these clusters tended to sum to an overall frequency significantly greater or less than 1 in some samples, suggesting that the strain frequency pipeline was failing to accurately capture all strains or spuriously inferring too many. As a result, we excluded species × host pairs with three strain clusters and limited our analysis to species × host pairs that had two strains. At times, the strain inference algorithm outputs two clusters, which is inconsistent with a scenario of either 1, 2, or 3 strains colonizing and instead may be arising from high variance in allele frequency. This happened once in the humanized mouse data with *B. uniformis*, the strain inference algorithm inferred *two* clusters (Supplementary Fig. 15A). It was visually apparent in this case that the two clusters likely represented the same group of SNVs segregating between two strains' genetic backgrounds because of their highly correlated trajectories, but were separated into two distinct clusters due to there being a large variance in individual SNV frequency. In this singular instance, we reclustered by using the mean SNV frequency of all SNVs in both clusters as a focal SNV (Supplementary Fig. 15B) and extracting all other SNVs connected to it by an edge in the network (i.e., with $d < 3.5$). This method produced a single cluster. The algorithm was able to infer strain frequencies for all other species analyzed in this study from the humanized and conventional mouse datasets without the need for ad hoc modifications to the pipeline. For several species × host pairs in the Shalon et al., 2023 data, the strain inference algorithm yielded two clusters. These species × host pairs were excluded from downstream analysis of strain frequency in favor of species × host pairs with more unambiguous strain phasing results, all of which had two strains (i.e., a single cluster).

When applying this algorithm to the humanized mouse data, it in some instances inferred only single strain for a bacterial species across all samples despite the inoculum displaying high nucleotide diversity, suggestive of multiple strains being present in the inoculum. This phenomenon may occur when applying the algorithm to a collection of samples in which a strain is present in only one sample (*e.g.*, the inoculum) but absent in all others (*e.g.*, the mouse samples). As a result, the algorithm would be unable to detect correlations between the SNVs private to a single sample and therefore fail to detect that strain. We identified when this might be happening in the humanized mouse data by looking for species in which only a single strain was inferred (i.e., the algorithm produced no clusters), but which had at least 1000 SNVs in the inoculum that were polymorphic and had a read support of four reads for the minor allele. When this scenario occurred, we assumed two strains were present in the inoculum and inferred the strain frequency $f_{strain1, inoculum}$ of the first strain as the mean of the

allele frequencies of these polymorphic SNVs in the inoculum, and the strain frequency $f_{strain2, inoculum}$ of the second strain as $1 - f_{strain1, inoculum}$. We generated 95% confidence intervals by applying the same stationary bootstrapping approach to the polymorphic SNVs identified in the inoculum.

## Calculating change in major strain frequency $\Delta f$ between samples

The change in major strain frequency $\Delta f$ was calculated as the magnitude of the difference in frequencies of one of the strains between two samples. Because all samples with co-colonizing strain analyzed in this study harbored two strains, $\Delta f$ was the same regardless of whether strain 1 or 2 was used to calculate the frequency change. For the humanized and conventional mouse data, we calculated $\Delta f$ between samples for co-colonizing species present in at least at least two mice, at least two cages, and at least two gut regions. When visualizing the distribution of $\Delta f$ values, we categorized pairwise comparisons as being "within host", "between host, within cage", and "between host, between cage". For each species, we selected a single "within host" pair per mouse, a single "between host, within cage" pair per cage, and a single "between host, between cage" pair for the three possible cage-wise comparisons. For "within host" comparison within each host, we selected the pair of samples that maximized the distance between the two sampling sites, breaking ties by selecting the pair of samples with the sample that is furthest down the gut (i.e., closest to the colon). For between host comparisons (both within and between cage), we selected the pair of samples that represented the same gut region or two gut regions proximal to one another along the gut, breaking ties by selecting pairs of samples with the sample that is furthest down the gut (i.e., closest to the colon). For "between host, between cage" comparisons, we ensured that no mouse was involved in more than one cross-cage comparison to eliminate redundancy. For Shalon et al., 2023 data, we calculated $\Delta f$ between samples belonging to co-colonized species x host pairs if the species were present in at least two timepoints and at least two device types. When visualizing the distribution of $\Delta f$ values, we categorized pairwise comparisons as being "within timepoint, between gut region", "between consecutive timepoints", and "between host first and last timepoint". For each species x host pair, we selected a single "within timepoint" pair per timepoint, a single "between consecutive timepoints" pair for each consecutive pair of timepoints, and a single "between host first and last timepoint" pair. For "within timepoint" comparisons within each timepoint, we selected the sample pair that maximized the distance between device types (i.e., biasing towards device type 1 versus device type 4 comparisons, if available), breaking ties by selecting the sample pair with the maximum device type number (i.e., biasing towards pairs with device type 4). For between timepoint comparisons (both consecutive and first and last timepoints), the sample pair that minimized the distance between device types and was furthest along the gut was selected.

## Performing ANOVA of major strain frequencies

To quantify the variance in major strain frequency across samples, we applied an ANOVA approach. In the humanized mouse dataset, the major strain was designated as the strain at frequency $f > 0.5$ in the inoculum. In conventional mouse and Shalon et al., 2023 data, the major strain was designated as the strain at frequency $f > 0.5$ in a majority of samples. The centered log ratio (CLR) transformation was applied to major strain frequency in all samples in which that species was detected, a commonly used approach to remove compositional artifacts when applying non-compositional approaches such as ANOVA[85,86]. For each species, we built an ANOVA model using the Stats package in R. In the humanized and conventional mouse data, the model was specified with CLR-transformed major strain frequency as the dependent variable and cage, mouse, and gut region as the independent categorical variables, and was applied jointly to all samples

belonging to the same study. Variance explained was calculated as the sum of squares between groups corresponding to each variable divided by the sum of squares total. We applied several prevalence requirements to ensure that the amount of variance explained by each dependent variable was properly quantified. In humanized and conventional mouse data, we only performed ANOVA on species detected across at least two cages (with the species detectable in at least two mice in those cages), at least two mice (with the species detectable in at least two gut regions in those mice), and two gut regions (with the species detectable in these same gut regions in at least two mice).

## Identifying quasi-phaseable lineages

SNV frequency differences between samples can arise due to evolution or due to shifts in the frequency of co-colonizing strains, which are likely to harbor thousands of differing nucleotides between them. To distinguish between SNV frequency differences that arise due to evolution versus those arising from strain frequency shifts, we identified quasi-phaseable lineages using an approach developed in Garud et al.[23]. By inferring quasi-phaseable lineages, we then were able to confidently infer evolutionary changes.

Briefly, the approach works by identifying samples in which a particular species has a single dominant strain at $f \geq 0.8$, hereafter referred to as a quasi-phaseable sample. In such samples, the allele corresponding to the dominant strain can be inferred with statistical confidence. Specifically, the alleles with frequency $f \geq 0.8$ at genomic loci with coverage $D \geq 20$ are assigned to the quasi-phaseable sample. Only samples with a median coverage of 20× for the species of interest are processed through the quasi-phaseable pipeline. Employing such stringent coverage and allele frequency thresholds ensures that the incorrect allele is not inferred, as it is statistically improbable that sampling error would give rise to the incorrect allele[23].

We made an important alteration to the quasi-phaseable pipeline in its application to the humanized and conventional mouse data relative to its application in Garud et al.[23]. Normally, the quasi-phaseable pipeline designates a sample as having a quasi-phaseable strain if the number of intermediate frequency SNVs (i.e., SNVs with frequency $0.2 < f < 0.8$) which a species harbors within the sample are 10% or less than the number of SNV differences that are typically observed for that species between independent hosts. This heuristic arises from the assumption that independent hosts should harbor genetically distinct strains that have diverged over long evolutionary timescales, and is born out by empirical evidence[23]. The 10% threshold is calibrated independently for each species, as genetically distinct strains of one species are not necessarily expected to show the same level of nucleotide differentiation as genetically distinct strains of a different species. In previous applications of the quasi-phaseable pipeline, the number of SNV differences between independent hosts has been assessed from the dataset used in those studies. For example, Garud et al.[23] studied evolutionary dynamics within 693 unique individuals sampled as a part of the human microbiome project[87,88], the TwinsUK registry[89], a cohort of Chinese subjects[90], and four additional young twins[91]. Because most hosts included in this compiled dataset were unrelated (and in some cases, on different continents), the assumption that their resident strains were distantly diverged was a reasonable one. By contrast, the humanized and conventional mice are expected to have low between-host strain diversity, either because they were inoculated with the stool from a single human host or because their microbiomes were homogenized early on in development through co-housing. As a result, we could not accurately quantify the number of differences that are expected between two unrelated hosts, making it impossible to detect quasi-phaseable strains using this dataset alone. We took two different approaches to overcome this issue in humanized and conventional mice, respectively. In humanized mice, we calculated the expected number of between-host SNV

differences for each focal species based on the aforementioned data used in Garud et al.[23], which we had previously processed through MIDAS 1. Because this resource was not available for the conventional mouse data processed through MIDAS 3, we designated samples as quasi-phaseable if intermediate frequency sites (i.e., SNVs with frequency $0.2 < f < 0.8$) occurred at a rate of $1 \times 10^{-4}$ per bp or less, consistent with rates observed in quasi-phaseable samples from humanized mice. Results published in this study (Fig. 3A) and elsewhere[23] suggest that this is a highly conservative threshold for the maximum amount of intermediate frequency diversity we would expect to see if only a single strain were present within a host.

## Inferring SNV frequency changes

To detect SNV frequency changes arising as a result of evolution, we follow the approach in Garud et al., 2019. A SNV frequency change is defined as an allele with low frequency in one quasi-phaseable sample ($f \leq 0.2$) and high frequency in another quasi-phaseable sample ($f \geq 0.8$). SNV differences were only considered at loci that had a coverage $D \geq 20$ in both samples being compared. An extreme allele frequency change of this magnitude and this minimum depth is statistically unlikely to arise between quasi-phaseable strains due to sampling error or drift given large population sizes and short timescales considered in this study[23,37]. Pairs of samples harboring $\leq 20$ SNV differences were considered to have the same inoculum strain evolving over short timescales (e.g., the eight-week timescale considered in this study), as larger amounts of SNV differences, such as $\mathcal{O}\left(10^{3}\right)$ or greater are inconsistent with within-host rates of diversification[23,41,77,80]. Consequently, pairs of quasi-phaseable strains with $> 20$ SNV differences were excluded from this evolutionary analysis. For more information, see the Supplementary Information for Garud et al.[23].

## Bootstrapping confidence intervals for SNV difference rates

To measure whether there were significant differences in the number of SNV frequency changes occurring between samples within hosts, between hosts, and between the inoculum and mouse hosts in the humanized mouse data, we employed a bootstrap approach to compute confidence intervals for the rate of SNV difference between quasi-phaseable pairs of samples with the same dominant strain. In 1000 bootstrap iterations, we sampled 100 quasi-phaseable sample pairs (not necessarily of the same species) with replacement from each comparison category: within host, between host, and between inoculum and host. For the within-cage comparison category, we only considered quasi-phaseable pairs from different hosts. We computed the actual rate of SNV frequency changes for a particular comparison category as the total number of SNVs displaying extreme allele frequency changes across all sampled quasi-phaseable pairs divided by the total number of high coverage loci across all sampled quasi-phaseable pairs. We calculated 95% confidence intervals using the cumulative distribution function of the overall SNV change difference rate in each comparison category generated from the 1000 bootstrap iterations, taking the 2.5 and 97.5 percentile as the upper and lower bound, respectively (Fig. 5B). We applied an identical approach for assessing whether SNV frequency change rates were significantly different between samples from the same host or different hosts in the conventional mouse data (Fig. 6C).

## Statistics and reproducibility

No statistical method was used to predetermine sample size for the humanized mice and conventional mice study. Mice were assigned to cages based on sex, and the investigators were not blinded to allocation during experiments. The statistical analyses conducted for each section are described in the corresponding text. No data were excluded from these analyses. For the analyses of Shalon et al. data, no samples were excluded from these analyses.

## Data availability

The raw metagenomics sequencing reads data generated in this study have been deposited in the NCBI Sequence Read Archive (SRA) under BioProject accession number PRJNA1230553. The processed count tables, associated metadata, and other data tables used to produce figures are available in Supplementary Information.

## Code availability

All necessary metadata, as well as the source code for the sequencing pipeline, downstream analyses, and figure generation, are available on GitHub (https://github.com/garudlab/Wasney-Briscoe/).

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

## Acknowledgements

The authors thank Kristin Harper, PhD, of Harper Health & Science Communications, LLC, for providing editorial support in accordance with *Good Publication Practice* guidelines, and Alison Feder, PhD, for her helpful edits. The authors also thank the Garud and the Tropini labs for their valuable inputs. In particular, the authors thank Peter Laurin for his statistical advice and feedback on the manuscript. Finally, the authors thank Dr. Sidhartha Sinha and members of the Sinha Lab at Stanford University for helpful discussions. This work was funded by NIGMS NIH award R35GM151023 (to N.R.G.), NSF CAREER award (no. 2240098, to N.R.G), a Paul Allen Distinguished Investigator Award (to C.T. and N.R.G.), a Canadian Institutes of Health Research Team Grant: Canadian Microbiome Initiative 2 (to C.T.), Crohn's and Colitis Canada, Canadian Institute for Advanced Research (to C.T.), the Michael Smith Foundation for Health Research Scholar Award (18239, to C.T.), Canada Foundation for Innovation/Infrastructure Operating Fund (38277, to C.T.), Canada Tier 2 Research Chair, Quantitative Microbiota Biology for Health Applications (CRC-2022-00036, to C.T.), Canadian Institute for Advanced Research/Humans and the Microbiome (FL-001253 Appt 3362, to C.T.), and the 4-Year Fellowship (to H.G.).

## Author contributions

N.R.G. conceived of the study. M.W., L.B., and N.R.G planned all analyses. M.W. and L.B. completed analyses and wrote all code associated with this study, with code and conceptual contributions from R.W. for the strain frequency estimation analyses. H.G. completed the mouse experiments and DNA extraction. M.W., L.B., C.T, and N.R.G. wrote the manuscript. All authors approved the final version of the manuscript.

## Competing interests

The authors declare no competing interests.
