## [Transparent Peer Review file · Nature Communications]

Uniform bacterial genetic diversity along the gut

Corresponding Author: Dr Nandita Garud

Version 0:

Reviewer comments:

Reviewer #1

(Remarks to the Author)

In this manuscript, Wasney, Briscoe et al, aimed to explore the distribution of strains and evolutionary modifications along the gut and “the ecological and evolutionary processes that give rise to this organization”. For this, they inoculated 8 germ-free mice with human stool sample from a single healthy volunteer. After eight weeks, they collected luminal contents of five intestinal regions, subjected them to metagenomic sequencing and analyzed species abundances and strain and SNV frequencies. They answered the following questions:

How is the microbiota different along the gut?

They find a different composition of gut microbiota along the gut.

Does within-species genetic diversity vary along the gut?

Nucleotide diversity is lower within mice than between mice

Do strain frequencies vary along the gut?

Mouse and cage identity are more important for this than gut region

How are genetic variants distributed along the gut?

More SNV frequency changes between hosts than within hosts.

Major points:

1.
How valid is their experimental system to answer the underlying question? The authors want to address how genetic diversity within bacterial species distribute along the gut and “the ecological and evolutionary processes that give rise to this organization” (line 25).
First, they do not analyze ecological and evolutionary processes, only bacterial composition, strain diversity and genetic diversity. Evolutionary ecology studies how evolutionary and ecological processes influence each other. Factors like resource competition and environmental shifts drive evolutionary change. Using human microbiota in mice makes it hard to draw conclusions about eco-evolutionary adaptation in mice. To assess co-evolution with the mouse gut, it's essential to use a species-specific microbiota [1].
Second, they use a very artificial system by colonizing germ-free mice with human stool. As the microbiota is species-specific, it would be more suitable to use mouse microbiota in this system (as mentioned by the authors in the discussion). Therefore, the authors need to perform the same experiment with transplanting mouse microbiota or frame their conclusions in a way that is more applicable to their specific experimental setting.
Also, they use stool to colonize the whole gastrointestinal tract. How suitable is this? And how well does the colonization work? E.g. Lactobacillaceae seem to have a very low abundance in the inoculum but are present at high abundance in the small intestine of some mice.
2.
The authors should think about a way to display the data in Figure 3 in a way that is better understandable and that underlines their point of a higher nucleotide diversity between mice than within mice. Also, it is not clear whether they plot each gut region or separate by small intestine/large intestine.

3.
How well can these results be transferred to humans? As the authors mention in the discussion, mice have a coprophagic behavior and thus re-inoculate themselves with colon material while humans do not do this. E.g. a mutation occurs in the colon and would – in humans – usually not reach the upper gastrointestinal tract, which would happen in mice due to coprophagic behavior.

4.
The authors present only sequencing data, without any functional analyses such as metabolomics, which could reveal region-specific activity along the gastrointestinal tract. It is possible that strains and genetic mutations are evenly distributed throughout the gut (Line 33-34) which does not necessarily mean that the function is also the same.

Minor points:

Line 70: Incomplete sentence, verb is missing

Line 90-91: How long does it take for human microbiota to fully colonize the mouse gastrointestinal tract?

Lines 112–113: Why were outbred mice (Swiss Webster mice) used? According to the metaorganism theory [2], both the host genome and microbiome shape the organism's phenotype. Using outbred mice introduces genetic variability—so how can the study rule out that host genetics influenced the gut microbiome?

Line 147-151 and 154-155: The authors should show the data that corresponds to this.

Line 161: The PCoA also shows a segregation by cage/chow. The authors should comment on this

Line 182: Please show the data that support that “Diversity measured from the inoculum was highly correlated with diversity measured from the mouse gut averaged for each species”.

Line 269: “For example, for the species *A. equilofaciens*, which was present in multiple mice and cages, the within-cage variance of strain frequencies measured in the jejunum ranged from 1.65×10^{-6} in cage 3 to 1.29×10^{-3} in cage 2, whereas the between-cage variance was at least two orders of magnitude greater 1.18×10^{-1} (Figure 4).” Where do I see this?

Line 338: What are the 11 SNV frequency changes they are referring to? Which one showed spatial differentiation? Which are the 4 unique within-host SNV frequency changes? What are the 57 unique between-host SNV frequency changes?

Line 489: “nine” instead of “nice”?

The study uses 9-week-old mice for inoculation (Line 489). However, this overlooks ontogenetic effects—key developmental factors that influence microbial adaptation to specific gut regions. To account for this, mice should be inoculated with the human microbiota before breeding, and microbial composition should be analyzed in the F1 generation.

Line 496: We appreciate, that the authors want to “ascertain replicability and generalizability”. However, the different diet seems to make a huge difference (see PCoA and Ng et al., *mBio* (2023)). So, it would only be fair to include sex and diet as separate variables.

Line 704: Supplementary Figure 6 instead of Supplementary Figure 5?

References:

1.
Chung, H., et al., Gut Immune Maturation Depends on Colonization with a Host-Specific Microbiota. *Cell*, 2012. 149(7): p. 1578-1593.

2.
Stappenbeck, T.S. and H.W. Virgin, Accounting for reciprocal host-microbiome interactions in experimental science. *Nature*, 2016. 534(7606): p. 191-9.

(Remarks on code availability)

Reviewer #2

(Remarks to the Author)

The study by Wasney et al. provides a detailed gut biogeographical analysis of strain-level diversity in a humanised gnotobiotic mouse model. As stated by the authors, the main motivation was to “characterise how genetic diversity varies along the gut”. Overall, the study is careful, thoughtful, and precise, and the manuscript is clear and well written. However, the experimental design and scope of the sampling and controls is unfortunately quite limited, and perhaps best put by the weaknesses outlined in the Discussion by the authors themselves. If the main motivation was indeed to characterise how genetic diversity varies along the gut, the design and sampling would arguably be more thorough. And while the analysis

was carefully conducted and there are some noteworthy observations, I do question how much these limited observations should really be generalized. The picture is simply too premature.

Here are some more specifics.

1. As stated by the authors themselves, things might look very different if they had included native mouse microbiomes as a control / comparison group. As it stands, statements like page 14, lines 364-367:

“This fact would seem to suggest that even when SNVs are adaptive, they are either broadly adaptive to the entire gut, or alternatively have high migration rates that outweighs local selection pressures, perhaps due to mixing and peristalsis along the gut”

Could for example be attributed to adaptations to universal aspects of the drastically new mouse host gut environment. Without mouse samples as both controls and a necessary group comparison, it really limits what can be inferred here. I would recommend to gavage native mouse stool(s) into germ free mice as an appropriate control/comparison.

2. Only a single host inoculum was used. There could also be peculiarities of any given single sample. The conclusions would be much stronger if e.g. these general patterns could be observed using more than one inoculum.

3. Related to 2., it seems as if there is very high level of replication, i.e. eight recipients for just a single donor. However, it only becomes apparent in the methods section that there is a “cryptic experiment” within this experiment, as the “mice in cages 2 and 3 were exposed to a fiber-rich diet of 30% guar gum for two weeks, while the rest continued on the standard diet.” It is argued that this added to the generalizability of the study, but more human and mouse donor material would have arguably offered more genuine generalizability. Further, without longitudinal sampling, it is not clear whether this 2 week intervention had an effect on the results or not.

Thus, in summary, it regretfully appears that the design reflects a re-purposing of samples from a very unrelated previous study (ref 3), rather than making a dedicated attempt at addressing the actual stated topic.

Additional questions:

Lines 337-339, are any of the 4 unique within-host SNV frequency changes between the upper and lower GI tract?

Lines 339-341, is there any additional higher level evidence of parallelism among the 57 unique between-host SNV frequency changes, e.g. at the level of KEGG pathways of the genes involved?

(Remarks on code availability)

Reviewer #3

(Remarks to the Author)

(Remarks on code availability)

Version 1:

Reviewer comments:

Reviewer #1

(Remarks to the Author)

I would like to sincerely thank the authors for their additional efforts and for carefully addressing my questions and concerns. I am satisfied with the revised version of the manuscript and believe that it has improved substantially in terms of clarity, rigor, and overall quality.

(Remarks on code availability)

N/A

Reviewer #2

(Remarks to the Author)

The authors have made considerable and sufficient efforts in addressing my previous concerns, well done!

(Remarks on code availability)

Reviewer #3

(Remarks to the Author)

(Remarks on code availability)

Response to Reviewers

Reviewer #1 (Remarks to the Author)

In this manuscript, Wasney, Briscoe et al, aimed to explore the distribution of strains and evolutionary modifications along the gut and “the ecological and evolutionary processes that give rise to this organization”. For this, they inoculated 8 germ-free mice with human stool sample from a single healthy volunteer. After eight weeks, they collected luminal contents of five intestinal regions, subjected them to metagenomic sequencing and analyzed species abundances and strain and SNV frequencies. They answered the following questions:

How is the microbiota different along the gut?

They find a different composition of gut microbiota along the gut.

Does within-species genetic diversity vary along the gut?

Nucleotide diversity is lower within mice than between mice

Do strain frequencies vary along the gut?

Mouse and cage identity are more important for this than gut region

How are genetic variants distributed along the gut?

More SNV frequency changes between hosts than within hosts.

Major points:

1.

How valid is their experimental system to answer the underlying question? The authors want to address how genetic diversity within bacterial species distribute along the gut and “the ecological and evolutionary processes that give rise to this organization” (line 25).

First, they do not analyze ecological and evolutionary processes, only bacterial composition, strain diversity and genetic diversity. Evolutionary ecology studies how evolutionary and ecological processes influence each other. Factors like resource competition and environmental shifts drive evolutionary change. Using human microbiota in mice makes it hard to draw conclusions about eco-evolutionary adaptation in mice. To assess co-evolution with the mouse gut, it's essential to use a species-specific microbiota [1].

We thank the reviewer for their comments. First, to clarify our objectives, we do not attempt to study how evolution and ecology influence each other and agree with the reviewer that factors like resource competition and environmental shifts would potentially be important for such inquiries. To improve clarity on the objectives of our work, we have removed the mention “the ecological and evolutionary processes that give rise to this organization” and instead state “thus,

to be able to understand how the microbiome functions at a mechanistic level, it is essential to understand how genetic diversity is organized along the gut” (Lines 19-20).

Second, regarding the suitability of the microbiota used here, as we will explain further below, a major point in our revision was to analyze i) mice with their native, conventional microbiota, as well as ii) humans that have ingested a capsule that can sample luminal contents. In repeating our experiment, we found that the same patterns of uniform strain frequency and spread of evolutionary adaptations along the gut hold in native microbiomes and across different animal hosts. While this study does not aim to characterize co-evolution between mice and their native gut microbiota *per se*, we believe this addition does address confounders arising from not using a native host-microbiome pair. The results are addressed in the response to comment below and in **Figure 6**.

Second, they use a very artificial system by colonizing germ-free mice with human stool. As the microbiota is species-specific, it would be more suitable to use mouse microbiota in this system (as mentioned by the authors in the discussion). Therefore, the authors need to perform the same experiment with transplanting mouse microbiota or frame their conclusions in a way that is more applicable to their specific experimental setting.

We thank the reviewers for this important suggestion. To address this suggestion we repeated the experiment in conventional mice harboring native mouse microbes to confirm that the patterns seen in our study are not due to incompatibility effects between host and microbe. In this experiment, we analyze a cohort of nine fully conventional mice that were cohoused until weaning before being housed into three cages of three mice each. At six weeks of age, we sampled and sequenced luminal contents from the same five gut regions that were previously sampled in humanized mouse guts.

We describe our results in lines 418-434 of the paper:

“Further replicating our findings from humanized mice, we found that in conventional mice, co-colonizing strains of the same species exhibited largely uniform frequencies across gut regions. This pattern contrasts with the substantial variation in strain frequencies between mice, even if co-housed (**Figure 6A, Supplementary Figure 10**). Among the nine bacterial species with co-colonizing strains that were present in multiple mice, cages, and gut regions, median change in frequency Δf between gut segments within hosts was only 2.21×10^{-2} , much smaller than the median frequency changes observed between mice in the same cage (2.16×10^{-1}) and in different cages (1.42×10^{-1}) (**Figure 6B**). Corroborating these results, gut region explained the least variation (average variance explained = 0.75%), whereas mouse identity explained the most variation (average variance explained = 78.39%) (Supplementary Figure 11). Cage explained a moderate amount of variation (average variance explained = 19.97%). These results indicate that uniform frequencies of strains along the gut is not unique to humanized mice and may be a common feature of mouse microbiota.

Additionally, we did not observe evolutionary changes accruing between gut regions within hosts even while changes arose between hosts at a rate of 1.03×10^{-6} changes/bp (Figure 6C). These results indicate that any evolutionary changes arising within hosts sweep globally throughout the gut.”

We show Figure 6A-C in Figure R1 below:

Figure R1 (Figure 6A-C in the manuscript). Uniform genetic variation along the gut in conventional mice and healthy humans. (A) Strain frequencies for two co-colonizing strains of an Oscillospiraceae species long the guts of nine conventional mice housed across three cages. Error bars represent bootstrapped 95% confidence intervals. Mice 1-3 were co-housed in cage 1, mice 4-6 in cage 2, and mice 7-9 in cage 3, as indicated by the light and dark purple boxes. Absence of a strain frequency bar indicates insufficient coverage or species absence in a given sample. For additional strain frequency plots corresponding to species analyzed in (B), see Supplementary Figure 10. **(B)** Change in strain frequency Δf was calculated for seven species present in multiple mice, cages, and gut regions. (Methods). Each dot represents the change in major strain frequency between a pair of samples from within the same host (red), between hosts in the same cage (light purple), and between hosts in the different cages (dark purple). **(C) Comparison of the rate of SNV frequency changes within and between mice. Colored in red are the observed SNV frequency change rates aggregated across species. Bars represent bootstrapped 95% CIs.**

To assess generalizability of our findings beyond mice, we also tested our hypothesis in humans. We analyzed previously published metagenomic data collected from 15 people over multiple time points using ingestible capsules designed to sample luminal contents from four successive regions of the gastrointestinal tract: pyloric sphincter to the upper small intestine (device type 1), upper to mid-small intestine (device type 2), mid- to lower small intestine (device type 3) and the lower small intestine into the ascending colon (device type 4) (Shalon *et al.*, 2023).

We found that in humans, the frequency of co-colonizing strains generally remained homogenous along the four sampled regions of the gut. In some cases, we observed short-lived changes in strain frequency with time but these changes did not persist to the next time point. We also observed a paucity of SNV frequency differences along the gut, and those that we did detect were transient from one timepoint to the next, suggesting they arose through rapid, stochastic fluctuations rather than environmental selection along the gut. We describe these results in lines 435-478, which correspond to **Figure 6D-E (Figure R3 in the response) and Supplementary figure 12 and 15.**

Our results suggest spatial uniformity in genetic diversity may be a robust feature of mammalian gut microbiota. Importantly, the humanized mouse data is still crucial for the message of our work. By having an inoculum to compare to, we can ascertain if a lack of SNV differences along the gut is due to the absence of evolutionary changes accruing, or the global spread of an evolutionary change throughout the gut. Additionally, the replication across mice and cages allows us to understand the statistical robustness of our results. Humanized mouse models are standard models in human microbiome studies, and have been leveraged in several landmark studies to understand microbiome dynamics during bacterial colonization and community assembly (Seedorf *et al.*, 2014; Johansson *et al.*, 2015), as well as in the context of human infectious and inflammatory diseases (Sonnenberg *et al.*, 2016) and obesity (Turnbaugh *et al.*, 2009). For this reason, humanized mice serve as an important and more tightly controlled experimental model with which to explore the spatial organization of genetic diversity in the mammalian gut. We now cite these papers in the methods and have clarified our reasoning in the main text (lines 67-76).

Also, they use stool to colonize the whole gastrointestinal tract. How suitable is this? And how well does the colonization work? E.g. Lactobacillaceae seem to have a very low abundance in the inoculum but are present at high abundance in the small intestine of some mice.

Several landmark studies (Turnbaugh *et al.*, 2009; Seedorf *et al.*, 2014; Johansson *et al.*, 2015; Sonnenberg *et al.*, 2016) have used stool for inoculation of humanized mice. Additionally, due to the coprophagic nature of mice, it is not unusual for stool-associated microbes to pass through and potentially colonize the whole GI tract. One limitation of using stool as an inoculum is that small intestinal oxygen intolerant microbes, such as Lactobacillus, could be missing from these samples. We address this concern by evaluating the abundance of Lactobacillus in the small intestine in our study. As expected, we observed Lactobacillus to have high abundance in the small intestine and lower abundance in the large intestine.

We added a paragraph addressing whether small intestine-associated bacteria colonized the small intestine in the species section (lines 146-153):

“Meanwhile, the family Lactobacillaceae, which is known to colonize the human small intestine with the aid of unique mucus-binding proteins (MUBs) which adhere to the small intestinal mucosal layer²⁸, was enriched in the small intestine relative to the large intestine (median log2 fold change per mouse = -0.121; Wilcoxon signed-rank test, p-value = 0.0225) (**Supplementary Figure 1**). These enrichments in the small intestine confirm that despite human stool more closely resembling the community composition of the human large intestine²⁹, inoculation with a human fecal sample is sufficient to support colonization by small intestine-associated taxa.”

Note that **Supplementary Figure 1** is included as **Figure R4** in this response.

2. The authors should think about a way to display the data in Figure 3 in a way that is better understandable and that underlines their point of a higher nucleotide diversity between mice than within mice. Also, it is not clear whether they plot each gut region or separate by small intestine/large intestine.

We thank the reviewer for these comments and the suggestions have improved the paper. To address the reviewers comments, Figure 3 has been modified in the following ways:

- First, to more directly compute genetic differentiation between samples, an additional figure (**Figure 2B**) has been added to show fixation index F_{ST} , or genetic distance, calculated between samples within the same host vs different hosts. This statistic was useful for demonstrating that subpopulation genetic differentiation is smaller within hosts than between hosts, particularly for species with high nucleotide diversity in the inoculum, indicating that the frequencies and identities of colonizing strains are more similar within than between hosts.
- Additionally, in **Figure 2A**, different gut segments have now been given different shapes to make it clear that we are plotting all gut segments as opposed to just small versus large intestine. In the figure caption we make clear that all gut segments are plotted.

The text has been updated as follows (lines 202-211):

“To understand if nucleotide diversity differed between regions of the gut, we measured F_{ST} , or fixation index, between the small and large intestine for each species. F_{ST} is a distance metric where high values indicate large genetic differentiation between two samples. We hypothesized that F_{ST} would be elevated between gut regions within hosts, either due to evolutionary changes accruing or variable strain frequencies. Contrary to this expectation, we observed that F_{ST} between the small and large intestine of the same host was on average low for the 30 species examined (0.119), especially relative to the average F_{ST} between mice at the same gut region (0.225) (**Figure 3B**). This difference was more pronounced in species with high nucleotide diversity in the inoculum ($\pi > 10^{-3}$)

(Wilcoxon rank sum test, p -value = 2.3×10^{-7}) compared to species with low diversity ($\pi \leq 10^{-3}$) (Wilcoxon rank sum test, p -value=0.043).”

We include **Figure 3B** as **Figure R2** below:

Figure R2 (Figure 3 in the manuscript). Genetic diversity measured in 30 gut commensal species. (A) Nucleotide diversity (π) estimates for the 30 most abundant and prevalent species (**Methods**). Each dot represents nucleotide diversity (π) in a specific gut region of an individual mouse. Dot color corresponds to the mouse, while dot shape indicates the gut region. Red

asterisks denote nucleotide diversity measured in the inoculum for that species. Species names in light red correspond to those species with $\pi > 1 \times 10^{-3}$ in the inoculum. **(B)** Fixation index (F_{ST}) for between all pairs of mouse samples for the 30 species analyzed in panel **(A)**. Each data point represents a single species for a given pair of samples, with points colored according to whether the sample comparison is within host (grey) or between hosts (black). Data points are further separated based on nucleotide diversity of the species in the inoculum. Two separate Wilcoxon rank sum tests were performed comparing the distributions of F_{ST} within versus between hosts for species with $\pi > 10^{-3}$ and species with $\pi \leq 10^{-3}$, respectively (* indicates $p \leq 0.05$; **** indicates $p \leq 0.0001$).

3.

How well can these results be transferred to humans? As the authors mention in the discussion, mice have a coprophagic behavior and thus re-inoculate themselves with colon material while humans do not do this. E.g. a mutation occurs in the colon and would – in humans – usually not reach the upper gastrointestinal tract, which would happen in mice due to coprophagic behavior.

We analyzed data from Shalon *et al.*, 2023 to assess if these results are applicable to humans. We found that strain frequency was uniform across gut regions and timepoints. Moreover, there were few SNV frequency differences along the gut, and those that we did detect tended to be transient, suggesting they arose through rapid and potentially stochastic temporal fluctuations rather than environmental selection. We have added the following to the text in lines 435-478:

“Next, to determine whether spatially uniform patterns of genetic diversity are a feature of human guts as well, we analyzed previously published metagenomic data⁵⁴ collected using ingestible sampling capsules designed to sample luminal contents from four successive regions of the gastrointestinal tract: pyloric sphincter to the upper small intestine, upper to mid-small intestine, mid- to lower small intestine and the lower small intestine into the ascending colon. In this dataset, 15 healthy adult subjects ingested four capsule types twice daily for as long as a week.

We hypothesized that strain frequencies in humans might vary more along the gut than in mice, given the greater length and compartmentalization of the human intestine, as well as the potential for host-specific niche adaptation over long residence times^{25,27}. Contrary to this expectation, strain frequencies were generally homogeneous not only along the gut but also across timepoints. For example, the frequency of the major strain of *Ruminococcus obeum* in Subject 11 fluctuated around a mean of 72% with a variance of 0.4% across gut locations and timepoints (**Figure 6D**). More generally, across the 25 species x host pairs with co-colonizing strains in multiple timepoints and device types, the median change in major strain frequency Δf between gut regions within the same timepoint was only 3.79×10^{-2} . This was the same order of magnitude as the median change observed between consecutive timepoints (2.92×10^{-2}), or even between the first and last observed timepoint for a species x host pair (3.55×10^{-2}) (**Figure 6E**). These results indicate that strain frequency tends to be uniform along the human gut as well as across hourly and daily timescales.

While the majority of species x host pairs never showed a temporal or spatial fluctuation in strain frequency exceeding 25%, there were several outlier species x host that harbored strains undergoing large frequency temporal shifts, and to a lesser degree, spatial shifts. However, these fluctuations tended to be transient. For example, the frequency of the major strain of *Burkholderiales bacterium* in Subject 6 fluctuated between a minimum of ~10% and maximum of ~95% within and across timepoints, displaying no consistency in the direction of change across space or time (**Supplementary Figure 12**). We observed this in other species x host pairs as well, illustrating that strain fluctuations that do occur along the gut and over time are short-lived, consistent with previous work showing that stably co-colonizing strains can fluctuate in frequency around a carrying capacity over longer timescales²⁵.

We next asked whether extreme SNV frequency changes arising as a consequence of within-host evolution were spatially segregated. To do so, we identified SNV frequency changes between distinct regions of the gut for each of the 56 host x species pairs (representing 35 species detected in 11 subjects) with enough data in at least two capsules for a given time point. Out of the 56 pairs, only 7 of these (12.5% of species x host pairs) had detectable SNV frequency changes between gut regions within a single timepoint (**Supplementary Table 6**), indicating that spatial segregation of SNVs is a feature in only a minority of samples.

Finally, we investigated whether the SNV changes arising along the gut are observable over multiple time points, or whether they are transient. Persistence would suggest that environmental gradients maintain spatial segregation of genetic variation along the gut over time. Of the seven host x species pairs with spatial differences, four had sufficient temporal and spatial data (at least two timepoints with at least two device types per timepoint). In three of four host x species pairs, spatial differences did not persist over multiple time points, with allele frequencies often reverting to their original state (**Supplementary Figure 13, Supplementary Table 6**). These results imply that sustained extreme allele frequency changes along the gut are rare.”

Figure 6D and E is included below as **Figure R3** to demonstrate the spatial uniformity in strain frequency that we observed in the Shalon *et al.*, 2023 data set. These results corroborate that the findings of our humanized mouse study can offer insight into the human microbiome.

Figure R3 (Figure 6D-E in the manuscript): (D) Strain frequencies for two co-colonizing strains of *Ruminococcus obeum* were inferred from capsule devices that sampled luminal contents along the gut of a healthy human subject (subject 11) across six timepoints. Device type 1 targeted the pyloric sphincter to the upper small intestine, device type 2 targeted the upper to mid-small intestine, device type 3 targeted the mid- to lower small intestine, and device type 4 targeted the lower small intestine into the ascending colon⁵⁴. Error bars represent bootstrapped 95% confidence intervals. Timepoints represent the time at which capsules were swallowed, with days being relative to the first timepoint plotted and times of day being coarsened into “morning” (before 12 pm PST), “afternoon” (after 12pm and before 8 pm PST), and “evening” (after 8 pm PST) bins. For additional strain frequency plots corresponding to species analyzed in (E), see **Supplementary Figure 12. (E)** Change in strain frequency Δf was calculated between samples belonging to 25 species x host pairs with co-colonizing strains present in multiple timepoints and device types. (**Methods**). Each dot represents the change in major strain frequency between samples from different device types swallowed during the same timepoint, (red), collected during consecutive timepoints (light purple), and collected during the first and last timepoint (dark purple). Strain frequency change over the course of the entire sampling period never exceeded changes occurring between consecutive timepoints, suggesting that hourly or daily shifts in strain frequency are transient.

Additionally, we include **Supplementary Figure 13** to illustrate the transience of local SNV changes along the guts of humans.

The authors present only sequencing data, without any functional analyses such as metabolomics, which could reveal region-specific activity along the gastrointestinal tract. It is possible that strains and genetic mutations are evenly distributed throughout the gut (Line 33-34) which does not necessarily mean that the function is also the same.

We thank the reviewer for this comment. We performed additional functional profiling on the metagenomic data from humanized mice to directly assess whether region-specific functional differences exist along the gastrointestinal tract. Using HUMAnN3 (Beghini et al., 2021), we examined the community level abundance of MetaCyc pathways and then applied MaAsLin3 (Nickols et al., 2024) to modeling abundance as a function of gut region, cage, and read depth as fixed effects with subject-level stratification for paired comparisons. This analysis revealed functional differentiation between the small and large intestine. In total, 80 pathways were significantly associated with gut location ($q < 0.05$), including 33 pathways with highly significant associations ($q < 0.001$; **Supplementary Figure 2, Supplementary Table 2**). When examining the taxonomic source of these pathways, the results were consistent with our findings for species diversity as the same families enriched in the large intestine—Bacteroidaceae and Rikenellaceae—were also the primary contributors to the pathways most strongly associated with the large intestine. Conversely, Lactobacillaceae, which was enriched in the small intestine, also contributed 8 pathways significantly associated with the small intestine and none associated with the large intestine. Together, these results demonstrate that even though strain-level and gene-level diversity is relatively uniform along the gut, functional activity is not. Instead, we observe clear region-specific functional signatures that align with the species compositional differences we report. These new analyses suggest that uniform genetic diversity does not imply uniform functional potential.

We now report these results in lines 155-168 of the paper.

Minor points:

Line 70: Incomplete sentence, verb is missing

Thank you for this correction, we have updated the text.

Line 90-91: How long does it take for human microbiota to fully colonize the mouse gastrointestinal tract?

Human microbiota samples can rapidly colonize the mouse gastrointestinal tract, with studies showing that transplanted human fecal communities establish within gnotobiotic mice in 3 to 7 days (Seedorf *et al.*, 2014). A different study found that mice gavaged with conventional mouse microbiota required about 2 weeks to develop a bacterial composition resembling the donor microbiota, with immune-associated changes appearing at week 4 and full intestinal mucus layer formation by week 6 (Johansson *et al.*, 2015). We therefore allowed a total of eight weeks

for the microbiota and host responses to equilibrate before sampling. We now cite these papers in the methods.

Lines 112–113: Why were outbred mice (Swiss Webster mice) used? According to the metaorganism theory [2], both the host genome and microbiome shape the organism's phenotype. Using outbred mice introduces genetic variability—so how can the study rule out that host genetics influenced the gut microbiome?

We believe the consistent gut colonization patterns seen across genetically distinct mice as well as unrelated humans is a strength of this study, as it allows us to rule out any kind of spatial patterns being attributed to host genotype.

Line 147-151 and 154-155: The authors should show the data that corresponds to this.

We have added **Figure R4** as **Supplementary Figure 1** in our manuscript to visually illustrate the median log₂ fold change in family relative abundance when comparing the large versus small intestine in the same host (**Methods**). Positive values indicate enrichment in the large intestine, while negative values indicate enrichment in the small intestine. We now reference **Supplementary Figure 1** in the portion of the results section describing differential taxonomic abundance along the guts of humanized mice (lines 139-150), and describe how these log₂ fold change values are calculated in the methods section (lines 745-754).

Figure R4 (Supplementary Figure 1 in the paper). Families have differential relative abundance along the gut. A paired Wilcoxon signed-rank test was used to estimate median log₂ fold change between paired family relative abundance values corresponding to the large and small intestine of the same mice (**Methods**). Taxa labels represent Order; Family, and are ordered from highest to lowest log₂ fold change. Positive log₂ fold change values indicate families that are enriched in the large intestine, whereas negative log₂ fold change values indicate families that are enriched in the small intestine.

Line 161: The PCoA also shows a segregation by cage/chow. The authors should comment on this

In our study design, diet and cage were necessarily confounded due to standard animal husbandry practices, which require co-housing of mice by diet to reduce stress given they are social animals. As such, the PCoA plot, created using species abundances, did reveal segregation by cage and diet, though largely driven by diet, consistent with previous literature showing that diet can significantly influence microbial community composition (Turnbaugh *et al.*, 2009). However, at the strain and evolutionary level, there was no effect of diet on the pattern of spatial uniformity observed along the gut. As such, since there was no contribution of diet to our main conclusions, we chose to remove this plot from the manuscript. We appreciate the reviewer's attention to this point and have noted the potential influence of cage and chow in the methods section.

Line 182: Please show the data that support that "Diversity measured from the inoculum was highly correlated with diversity measured from the mouse gut averaged for each species".

We have now included **Supplementary Figure 3 (Figure R6)** to show the positive correlation between nucleotide diversity in the inoculum and average nucleotide diversity observed across the mouse samples:

Figure R5 (Supplementary Figure 4 in the manuscript). Inoculum π and average mouse π . Nucleotide diversity (π) observed in the inoculum (x axis) and average nucleotide diversity observed across mouse samples (y axis) is visualized for the 30 most abundant and prevalent species in the humanized mouse cohort (**Methods**). Error bars represent the standard deviation of π values observed in mouse samples. The dashed grey line represents the identity line ($y = x$).

Line 269: “For example, for the species *A. equilofaciens*, which was present in multiple mice and cages, the within-cage variance of strain frequencies measured in the jejunum ranged from 1.65×10^{-6} in cage 3 to 1.29×10^{-3} in cage 2, whereas the between-cage variance was at least two orders of magnitude greater 1.18×10^{-1} (Figure 4).” Where do I see this?

We have added a new **Supplementary Table 4** to our supplement that supplies the variance statistics for between cage and within cage comparisons and have updated the text accordingly in Line 286.

Additionally, we have added **Supplementary Figure 7** showing the comparison of between cage variance and within cage variance, included below as **Figure R6**.

Figure R6 (Supplementary Figure 7 in the manuscript). Variance in strain frequencies within and between cages. Variance in major strain frequency was measured for samples belonging to the same gut region in different mice within the same cage (“Within cage,” dark orange) and between different cages (“Between cage,” light orange). Within cage bars represent the average within cage variance across the three cages and five gut regions, while between cage bars represent between cage variance averaged across the five gut regions.

Line 338: What are the 11 SNV frequency changes they are referring to? Which one showed spatial differentiation? Which are the 4 unique within-host SNV frequency changes? What are the 57 unique between-host SNV frequency changes?

We have updated **Supplementary Table 5** (formerly **Supplementary Table 3**) to provide more detail on the SNVs considered in this study, including the 11 mentioned here. **Supplementary Table 5** is a table containing the allele frequencies of all SNVs that underwent an extreme allele frequency shifts between a pair of samples, and includes indication of which sample comparison the change arose in. For clarity, we have added a column “Observed in” that identifies what kind of comparison the SNV frequency change was identified in. For example, the 11 SNV frequency changes mentioned in the text are referring to SNVs in which there was a large frequency shift

in the mice relative to the inoculum, so these can be identified by looking for SNVs in rows labeled with “Inoculum” in the “Observed in” column. Of these 11, one SNV in *Coprococcus* sp. showed a large rise in frequency from small intestine to large intestine (we have now noted in the text that this SNV was found in *Coprococcus*).

Overall, there were four SNVs in our dataset that showed an extreme allele frequency change along the gut within a host, including the SNV in *Coprococcus* sp. along with one SNV in *Enterococcus faecium* and two in *Ruminococcus* sp. The extreme within-host changes were exclusively observed between small intestinal and large intestinal gut regions. Their exact frequencies can be found by looking for SNVs labeled with “Within Host” in the “Observed in” column in **Supplementary Table 5**. For additional clarity on these 4 SNVs, we have added a tab to **Supplementary table 5** that describes the average frequencies of these SNVs in the small intestine (duodenum, jejunum, and ileum) and large intestine (cecum and colon) in the mouse in which the extreme frequency change was detected.

The frequencies of all SNVs detected in the humanized mouse microbiomes are visualized in **Figure 5** and **Supplementary Figure 8**.

Line 489: “nine” instead of “nice”?

Thank you for this correction, we have updated the text.

The study uses 9-week-old mice for inoculation (Line 489). However, this overlooks ontogenetic effects—key developmental factors that influence microbial adaptation to specific gut regions. To account for this, mice should be inoculated with the human microbiota before breeding, and microbial composition should be analyzed in the F1 generation.

We thank the reviewer for raising this point. While we agree that ontogenetic factors can shape microbiome-host interactions, our study's primary focus was not on host development but rather on the spatial distribution of microbial genetic diversity across the gut.

To mitigate concerns about adaptation timescales, our germ-free mice were colonized with human microbiota and equilibrated for eight weeks prior to analysis, a practice standard in the field (Turnbaugh *et al.*, 2009; Seedorf *et al.*, 2014; Johansson *et al.*, 2015; Sonnenberg *et al.*, 2016). This equilibration period allowed for stabilization of community structure and recovery following oral gavage.

To further address the reviewer's concern and assess whether our findings were specific to humanization, we performed an additional experiment using conventional mice with naturally acquired, host-adapted microbiomes. These mice were sourced from the same litter and co-housed until weaning to ensure shared microbial exposure and strain pools. At six weeks of age, we collected luminal contents from the same five gut segments sampled in our original study design. Remarkably, we again observed spatial uniformity in genetic diversity, suggesting that this pattern is not an artifact of host-microbiome mismatch or colonization timing, but rather a potentially generalizable feature of microbial diversity in the gut.

Line 496: We appreciate, that the authors want to “ascertain replicability and generalizability”. However, the different diet seems to make a huge difference (see PCoA and Ng et al., mBio (2023)). So, it would only be fair to include sex and diet as separate variables.

We thank the reviewer for this important point. In our study design, sex, diet, and cage were necessarily confounded due to standard animal husbandry practices, which require co-housing of mice by sex and by diet. This constraint made it statistically underpowered and biologically confounded, preventing us from including these variables independently in our models.

Rather than focusing on specific contributions of sex or diet, which we acknowledge can influence the microbiome, our goal was to examine whether the *pattern of spatial uniformity*—the central finding of our study—persisted across multiple conditions. Importantly, we observed that this spatial pattern holds true regardless of sex and diet. Moreover, we found the same trend in conventional mice, all of whom were male and fed the same diet of standard rodent chow. This indicates that sex and diet are likely not driving the spatial signal we are attempting to highlight in this study across various dataset.

We appreciate the reviewer’s attention to this point and have noted the potential influence of cage and chow in the methods section (lines 607-610):

“With this experimental set up, we were able to ascertain replicability and generalizability of the genetic homogeneity observed along the gut in three independent cages with varying diets and sexes, as well as contrast within- and across-host dynamics (**Figure 1**).”

Line 704: Supplementary Figure 6 instead of Supplementary Figure 5?

Thank you for this correction, we have updated the text.

Reviewer #2 (Remarks to the Author)

The study by Wasney et al. provides a detailed gut biogeographical analysis of strain-level diversity in a humanised gnotobiotic mouse model. As stated by the authors, the main motivation was to “characterise how genetic diversity varies along the gut”. Overall, the study is careful, thoughtful, and precise, and the manuscript is clear and well written. However, the experimental design and scope of the sampling and controls is unfortunately quite limited, and perhaps best put by the weaknesses outlined in the Discussion by the authors themselves. If the main motivation was indeed to characterise how genetic diversity varies along the gut, the design and sampling would arguably be more thorough. And while the analysis was carefully conducted and there are some noteworthy observations, I do question how much these limited observations should really be generalized. The picture is simply too premature.

Here are some more specifics.

1. As stated by the authors themselves, things might look very different if they had included native mouse microbiomes as a control / comparison group. As it stands, statements like page 14, lines 364-367:

“This fact would seem to suggest that even when SNVs are adaptive, they are either broadly adaptive to the entire gut, or alternatively have high migration rates that outweighs local selection pressures, perhaps due to mixing and peristalsis along the gut”

Could for example be attributed to adaptations to universal aspects of the drastically new mouse host gut environment. Without mouse samples as both controls and a necessary group comparison, it really limits what can be inferred here. I would recommend to gavage native mouse stool(s) into germ free mice as an appropriate control/comparison.

We thank the reviewer for these comments and have added two new major analyses to the paper to address them. First, we repeated the experiment in (i) conventional mice with native gut microbiota as well as (ii) analyzed humans whose luminal contents had been sampled over several spatial points. We found that spatial uniformity extends to these two systems as well. Specifically, we observed spatial uniformity in strain frequencies along the gut, and zero local genetic adaptations arising in distinct gut segments. These results indicate that the uniformity is not an artifact of the humanized mouse system, and instead is a generalizable feature of conventional mouse microbiomes. For more details on the results stemming from our conventional mouse and healthy human cohorts, see **Figure 6 (Figure R1, Figure R3), Supplementary Figure 9-13**, as well as our response to reviewer 1's comments on pages 4-6 and 9-11.

2. Only a single host inoculum was used. There could also be peculiarities of any given single sample. The conclusions would be much stronger if e.g. these general patterns could be observed using more than one inoculum.

By observing the same patterns of uniformity in conventional mouse microbiomes and in healthy humans, we can rule out that these results are an artifact of the peculiarities of any given sample.

3. Related to 2., it seems as if there is very high level of replication, i.e. eight recipients for just a single donor. However, it only becomes apparent in the methods section that there is a “cryptic experiment” within this experiment, as the “mice in cages 2 and 3 were exposed to a fiber-rich diet of 30% guar gum for two weeks, while the rest continued on the standard diet.” It is argued that this added to the generalizability of the study, but more human and mouse donor material would have arguably offered more genuine generalizability. Further, without longitudinal sampling, it is not clear whether this 2 week intervention had an effect on the results or not.

Thus, in summary, it regretfully appears that the design reflects a re-purposing of samples from a very unrelated previous study (ref 3), rather than making a dedicated attempt at addressing the actual stated topic.

We appreciate these comments and towards this end, we hope that we have ameliorated concerns about the generalizability of our claims by including results from 9 conventional mice (all the same sex and on a standard rodent diet) with natural microbiomes, as well as 15 humans of varying sexes and genetic backgrounds in our analysis.

The following descriptions of the conventional mouse experimental setup has been added to the main text to accompany **Figure 6A-C** and **Supplementary Figure 9** (Lines 407-417):

“Having established that uniform strain frequencies and spread of evolutionary adaptations along the gut is a common feature in humanized mice, we next investigated whether these same patterns also arise in hosts with established native microbiota, specifically conventional mice and healthy humans.

We first analyzed a cohort of nine conventional mice that were cohoused until weaning at three weeks of age before being housed into three cages of three mice each. At six weeks of age, we sampled and sequenced luminal contents from the same five gut regions that were previously examined in humanized mouse guts (**Methods**). Large intestinal samples harbored higher alpha diversity and a different family-level composition than small intestinal samples (**Methods**) (**Supplementary Figure 9**), confirming that the expected spatial organization does exist at higher taxonomic levels along the guts of the conventional mice.”

From these 9 subjects, we inferred that strain frequencies remained generally consistent along the gut. The general trend observed across species was quantified with ANOVA as described in **Figure 6A-B** and the accompanying main text (lines 418-430):

“Further replicating our findings from humanized mice, we found that in conventional mice, co-colonizing strains of the same species exhibited largely uniform frequencies across gut regions. This pattern contrasts with the substantial variation in strain frequencies between mice, even if co-housed (**Figure 6A**, **Supplementary Figure 10**). Among the nine bacterial species with co-colonizing strains that were present in multiple mice, cages, and gut regions, median change in frequency Δf between gut segments within hosts was only 2.21×10^{-2} , much smaller than the median frequency changes observed between mice in the same cage (2.16×10^{-1}) and in different cages (1.42×10^{-1}) (**Figure 6B**). Corroborating these results, gut region explained the least variation (average variance explained = 0.75%), whereas mouse identity explained the most variation (average variance explained = 78.39%) (**Supplementary Figure 11**). Cage explained a moderate amount of variation (average variance explained = 19.97%). These results indicate that uniform frequencies of strains along the gut is not unique to humanized mice and may be a common feature of mouse microbiota.”

Next, we asked whether SNV changes spatially segregated along the guts of conventional mice, unlike SNVs in our humanized mouse system. To the contrary, we found even fewer extreme

within-host SNV changes in our conventional mice than in our humanized mice, suggesting that genetic variation in natural mice is also uniformly distributed along the gut. We added the following passage in the manuscript to present these results (Lines 431-434).

“Additionally, we did not observe evolutionary changes accruing between gut regions within hosts even while changes arose between hosts at a rate of 4.73×10^{-7} changes/bp (**Figure 6C**). These results indicate that any evolutionary changes arising within hosts sweep globally throughout the gut.”

The following descriptions of the human study (Shalon *et al.*, 2023) in the main text accompanying **Figure 6D-E** are as follows (lines 435-453):

“Next, to determine whether spatially uniform patterns of genetic diversity are a feature of human guts as well, we analyzed previously published metagenomic data⁵⁴ collected using ingestible sampling capsules designed to sample luminal contents from four successive regions of the gastrointestinal tract: pyloric sphincter to the upper small intestine, upper to mid-small intestine, mid- to lower small intestine and the lower small intestine into the ascending colon. In this dataset, 15 healthy adult subjects ingested four capsule types twice daily for as long as a week.

We hypothesized that strain frequencies in humans might vary more along the gut than in mice, given the greater length and compartmentalization of the human intestine, as well as the potential for host-specific niche adaptation over long residence times^{25,27}. Contrary to this expectation, strain frequencies were generally homogeneous not only along the gut but also across timepoints. For example, the frequency of the major strain of *Ruminococcus obeum* in Subject 11 fluctuated around a mean of 72% with a variance of 0.4% across gut locations and timepoints (**Figure 6D**). More generally, across the 25 species x host pairs with co-colonizing strains in multiple timepoints and device types, the median change in major strain frequency Δf between gut regions within the same timepoint was only 3.79×10^{-2} . This was the same order of magnitude as the median change observed between consecutive timepoints (2.92×10^{-2}), or even between the first and last observed timepoint for a species x host pair (3.55×10^{-2}) (**Figure 6E**). These results indicate that strain frequency tends to be uniform along the human gut as well as across hourly and daily timescales.”

We then analyzed gut region-specific evolutionary changes in the human gut (Lines 464-478)

“We next asked whether extreme SNV frequency changes arising as a consequence of within-host evolution were spatially segregated. To do so, we identified SNV frequency changes between distinct regions of the gut for each of the 56 host x species pairs (representing 35 species detected in 11 subjects) with enough data in at least two capsules for a given time point. Out of the 56 pairs, only 7 of these (12.5% of species x host pairs) had detectable SNV frequency changes between gut regions within a single

timepoint (**Supplementary Table 6**), indicating that spatial segregation of SNVs is a feature in only a minority of samples.

Finally, we investigated whether the SNV changes arising along the gut are observable over multiple time points, or whether they are transient. Persistence would suggest that environmental gradients maintain spatial segregation of genetic variation along the gut over time. Of the seven host x species pairs with spatial differences, four had sufficient temporal and spatial data (at least two timepoints with at least two device types per timepoint). In three of four host x species pairs, spatial differences did not persist over multiple time points, with allele frequencies often reverting to their original state (**Supplementary Figure 13, Supplementary Table 6**). These results imply that sustained extreme allele frequency changes along the gut are rare.”

Figure 6 has been included in this response as **Figure R1** and **Figure R3**; **Supplementary Figure 9-13**, and **Supplementary Table 6** can be found in the Supplementary Information file.

With these additional studies we find that the tendency for co-colonizing strains to maintain uniform frequencies along the gut is not just unique to humanized mice but can be generalized to conventional mice and native human microbiomes. Moreover, local SNV frequency changes do not occur at all along the guts of conventional mice, and arise only transiently in humans, suggesting they do not represent spatial differentiation along the gut driven by environmental selection.

Additional questions:

Lines 337-339, are any of the 4 unique within-host SNV frequency changes between the upper and lower GI tract?

Yes, for all 4 unique within-host SNV frequency changes, the frequency changed between the upper and lower GI tract in at least one of the mice. All SNV frequency changes are shown in **Supplementary Table 5** along with a secondary sheet indicating the 4 changes that occurred between the small and large intestine. We also clarify this observation in the text as follows (lines 357-359): “we observed only 4 unique within-host SNV frequency changes in 3 out of the 33 species, all between the small intestine and large intestine, including the SNV in *Coprococcus* sp. (**Supplementary Table 4**).”

Lines 339-341, is there any additional higher level evidence of parallelism among the 57 unique between-host SNV frequency changes, e.g. at the level of KEGG pathways of the genes involved?

Yes, we found evidence of parallelism among the between-host SNV frequency changes. In all, these 57 observed changes occurred in 36 unique genes. Of those genes, 16 of them could be annotated with KEGG pathways in prokaryotes using an HMMER search with Kofam-KOALA (Aramaki et al. 2020). We found 8 of these genes involved in metabolism, 3 in genetic

information processing, and 3 in signaling. Of the 8 metabolism genes, 3 were explicitly involved with glycan biosynthesis and metabolism.

Reviewer #3 (Remarks to the Author)

This email has been sent through the Springer Nature Tracking System NY-610A-NPG&MTS

References

Aramaki, T. et al. KofamKOALA: KEGG Ortholog assignment based on profile HMM and adaptive score threshold. *Bioinformatics* 36, 2251–2252 (2020).

Beghini, F. et al. Integrating taxonomic, functional, and strain-level profiling of diverse microbial communities with bioBakery 3. *eLife* 10, e65088 (2021).

Chung, H., et al., Gut Immune Maturation Depends on Colonization with a Host-Specific Microbiota. *Cell*, 2012. 149(7): p. 1578-1593.

Eng, A. & Borenstein, E. Taxa-function robustness in microbial communities. *Microbiome* 6, 45 (2018).

Garud, N. R., Good, B. H., Hallatschek, O. & Pollard, K. S. Evolutionary dynamics of bacteria in the gut microbiome within and across hosts. *PLOS Biology* 17, e3000102 (2019).

Johansson, M. E. V. et al. Normalization of Host Intestinal Mucus Layers Requires Long-Term Microbial Colonization. *Cell Host & Microbe* 18, 582–592 (2015).

Nickols, W. A. et al. MaAsLin 3: Refining and extending generalized multivariable linear models for meta-omic association discovery. *bioRxiv* 2024.12.13.628459 (2024)
doi:10.1101/2024.12.13.628459.

Seedorf, H. et al. Bacteria from diverse habitats colonize and compete in the mouse gut. *Cell* 159, 253–266 (2014).

Shalon, D. et al. Profiling the human intestinal environment under physiological conditions. *Nature* 617, 581–591 (2023).

Sonnenburg, E. D. et al. Diet-induced extinction in the gut microbiota compounds over generations. *Nature* 529, 212–215 (2016).

Stappenbeck, T.S. and H.W. Virgin, Accounting for reciprocal host-microbiome interactions in experimental science. *Nature*, 2016. 534(7606): p. 191-9.

Turnbaugh, P. J. et al. The Effect of Diet on the Human Gut Microbiome: A Metagenomic Analysis in Humanized Gnotobiotic Mice. *Sci Transl Med* 1, 6ra14 (2009).

Vieira-Silva, S. et al. Species–function relationships shape ecological properties of the human gut microbiome. *Nat Microbiol* 1, 16088 (2016).